

# Indices of the Hadley circulation strength and associated circulation trends

Matic Pikovnik[1], Žiga Zaplotnik[1], Lina Boljka[2], and Nedjeljka Žagar[3]

[1]Faculty of Mathematics and Physics, University of Ljubljana, Ljubljana, 1000, Slovenia
[2]Geophysical Institute and Bjerknes Centre for Climate Research, University of Bergen, Bergen, 5020, Norway
[3]Meteorological Institute, Center for Earth System Research and Sustainability, Universität Hamburg, Hamburg, 20146, Germany

**Correspondence:** Žiga Zaplotnik (ziga.zaplotnik@fmf.uni-lj.si)

**Abstract.** This study compares the trends of Hadley cell (HC) strength using different HC measures applied to the ECMWF ERA5 and ERA-Interim reanalyses in the period 1979-2018. The HC strength is commonly evaluated by indices derived from the mass-weighted zonal-mean stream function. Other measures include the velocity potential and the vertical velocity. Six known measures of the HC strength are complemented by a measure of the average HC strength, obtained by averaging the stream function in the latitude-pressure ($\varphi$-$p$) plane, and by the total energy of unbalanced zonal-mean circulation in the normal-mode function decomposition. It is shown that measures of the HC strength, which rely on point values in the $\varphi$-$p$ plane, produce unreliable long-term trends of both the northern and southern HCs, especially in ERA-Interim; magnitudes and even the signs of trends depend on the choice of HC strength measure. The two new measures alleviate the vertical and meridional inhomogeneities of the trends in the HC strength. In both reanalyses, there is a positive trend in the total energy of zonal-mean unbalanced circulation. The average HC strength measure also shows a positive trend in ERA5 in both hemispheres, while the trend in ERA-Interim is insignificant.

## 1 Introduction

The Hadley circulation is a thermally forced overturning circulation, consisting of two symmetrical cells, which span between the tropics and the subtropics. Each cell consists of the ascending branch in the deep tropics, which is associated with enhanced precipitation, poleward upper-tropospheric flow, the descending motion in the subtropics that suppresses rainfall, and a frictional return flow in the lower troposphere. Therefore, potential changes of the Hadley cells (HCs), either to their strength or their meridional extent, will have a profound impact on the global hydrological cycle (Held and Soden, 2006; Burls and Fedorov, 2017) and the biosphere, particularly in the subtropics. For example, the subsidence region has already become drier because of the enhanced descending motion, in line with the satellite observations of upper tropospheric humidity and total water vapor (Sohn and Park, 2010).

A number of studies of the HC strength using reanalyses suggested strengthening of both the northern HC (NHC) and southern HC (SHC) in the recent decades. However, the reported magnitude and uncertainty of the trends differ (Tanaka et al., 2004; Mitas and Clement, 2005; Stachnik and Schumacher, 2011; Nguyen et al., 2013; Chemke and Polvani, 2019). This is,





alongside different reanalyses (with e.g. different resolutions) and study periods used, partly due to a variety of metrics that

have been used to define the HC strength. For example, the strength of the overall Hadley circulation can be evaluated using the velocity potential in the upper troposphere, e.g. at 200 hPa, as the meridional divergent flow in the upper branch of the HC is strongest there, which is associated with the maximal upward motions in the layer beneath (Tanaka et al., 2004). The Hadley circulation strength can also be defined by the minimum pressure velocity $\omega$ at some predefined mid-tropospheric level (Wang, 2002). Both measures describe the properties of the ascending branch of the Hadley circulation.

The majority of the studies describe the HC by the mass-weighted zonal-mean stream function $\psi$ in the latitude-pressure ($\varphi$-$p$) plane (Oort and Yienger, 1996). The $\psi$ function is computed by the vertical integration of the zonal-mean meridional wind

$$
\psi(\varphi, p) = \frac{2\pi R \cos\varphi}{g} \int\limits_0^p [v](\varphi, p') \mathrm{d}p',
\tag{1}
$$

where $[v]$ is the zonal- and annual/seasonal/monthly-mean meridional wind, $R$ is Earth's radius, $g$ is gravity, $\varphi$ is latitude and

$p$ is pressure. Several indices of the HC strength based on point values (maxima or minima) of $\psi(\varphi, p)$ have been used:

1. the maximum (minimum) values of $\psi$ in the $\varphi$-$p$ plane (e.g. Mitas and Clement, 2005; Stachnik and Schumacher, 2011; D'Agostino and Lionello, 2017);

2. the maximum (minimum) value of $\psi$ at some selected pressure level, e.g. 500 hPa (e.g. Kang et al., 2013; Son et al., 2018; Chemke and Polvani, 2019; Mathew and Kumar, 2019);

3. the vertical average of the maxima (minima) of $\psi$ at different pressure levels in the troposphere (e.g. in the layer 200 hPa - 900 hPa, as in Nguyen et al. 2013).

Nguyen et al. (2013) is also the only study that addresses the vertical inhomogeneity of the HC strength and its trends.

While several studies have compared the Hadley circulation in different reanalyses and climate models (e.g. Stachnik and Schumacher, 2011; Chemke and Polvani, 2019), no study (to our knowledge) has yet compared the measures of the HC strength

in the same dataset. In this study we perform such an inter-comparison and we assess how the trends estimated by different measures compare with each other in the ERA5 and ERA-Interim reanalyses. For example, we assess how sensitive are the trends derived from measures based on the latitude-pressure stream-function profile (1) to the choice of the pressure level. Motivated by uncertainties in the results based on the different measures, we propose two alternative measures of the HC strength: a) a stream-function based measure of average strength, which also grasps the overall trends of each Hadley cell; and

b) a normal-mode function based index which measures the strength of the global unbalanced zonal-mean circulation to which the Hadley cell makes the greatest contribution.

The paper is organised as follows. Section 2 describes the data and methods. The measures are compared in Section 3. Discussion and conclusions are given in Section 4.





## 2 Data and Methods

### 2.1 Reanalysis data

Two modern ECMWF reanalyses are analysed: ERA5 (Hersbach et al., 2020) and ERA-Interim (Dee et al., 2011). 40 years (1979-2018) of daily data at 00 UTC are used. Meridional wind ($v$), zonal wind ($u$) and vertical velocity ($\omega$) data are provided on 37 pressure levels between 50°S and 50°N on latitude-longitude grid with 1° resolution for both reanalyses. Among these levels, 23 are between 1000 hPa and 200 hPa.

The normal-mode function based index was computed using 40 years of monthly means of daily means for the zonal wind, meridional wind, temperature, geopotential and surface pressure. For details on the normal-mode function derivation and their applications, see Žagar et al. (2015) and Žagar and J. Tribbia (2020). For ERA5, the global data were analysed on regular Gaussian grid F80 with 1.125° resolution and 137 hybrid model levels. ERA-Interim data were analysed on the same horizontal grid but using 60 vertical levels.

### 2.2 Mean HC and its trend

Trends are evaluated from the time series of $\psi$ for different point values $\psi(\varphi, p)$ as linear regression coefficients. The trends are considered significant if they pass the 95% threshold of the modified Mann-Kendall test (Hamed and Ramachandra Rao, 1998). Note that the trends presented in this study are only representative of the analysed 40-year period and that we do not evaluate an extent to which they represent a climate-change signal. A separate study (Zaplotnik et al., 2021, in review) addresses this question and its results suggest that a part of the 40-year trends in the HC strength may be due to the multi-decadal variability.

Fig. 1 shows climatological monthly-mean stream function and its pointwise trends in the ERA5 reanalysis between 1979 and 2018. A significant enhancement of the winter cells can be observed: the northern Hadley cell (NHC; red contours) strengthens most between December and April, whereas the southern Hadley cell (SHC; blue contours) strengthens most between April and October. Both cells are strengthening between March and May. A prominent feature is that the trends in the monthly-mean HC strength are spatially inhomogeneous across the cells, both meridionally and vertically. For example, from December to February, the lower-tropospheric part of the descending branch in the NHC is strengthening, while the ascending branch of the cell in the deep tropics is weakening. From July to October, the SHC exhibits significant strengthening in the ascending branch in the Inter-Tropical Convergence Zone, while its descending branch mostly shows insignificant strengthening/weakening and even significant weakening on the southern boundary of the cell. The inhomogeneities in trends are even more pronounced in the ERA-Interim reanalysis (Fig. A1); for example, vertical inhomogeneity in the SHC trend from May to October is especially pronounced in the regions of the strongest $\psi$ gradients. The presence of the inhomogeneities in the HC trends raises a question about the reliability of some of the climate trends derived from point measures of the HC strength.

### 2.3 Measures of Hadley cell strength

The trends and their uncertainties are compared for several measures of the HC strength:





1. maximum (minimum) of annual/monthly-mean stream function between 40°S and 40°N and between 200 hPa and 900 hPa, denoted $\psi^{max}$ ($\psi^{min}$). Slightly different boundaries were employed by Mitas and Clement (2005); Stachnik and Schumacher (2011); D'Agostino and Lionello (2017); however, a reasonable choice of boundaries (e.g. excluding the lower part of the boundary layer and the stratosphere) does not affect the results;

2. maximum (minimum) of annual/monthly-mean stream function at predefined pressure levels (e.g. 400 hPa, 500 hPa, etc.) within 40°S and 40°N, denoted $\psi_p^{max}$ ($\psi_p^{min}$); as used in e.g. Kang et al. (2013); Chemke and Polvani (2019);

3. stream function value at the location of climatological (1979-2018) annual/monthly-mean maximum/minimum of the NHC/SHC strength, $\psi(\varphi^{max}, p^{max})$, where $(\varphi^{max}, p^{max}) = \underset{(\varphi, p)}{\operatorname{argmax}}(\overline{\psi_{1979-2018}})$, and analogous for $\psi(\varphi^{min}, p^{min})$;

4. an average of maximum/minimum values of annual/monthly-mean $\psi$ over pressure levels between e.g. 200 hPa and 900 hPa, with a constant step size of 50 hPa, as in Nguyen et al. (2013):

$$\langle \psi^{max} \rangle_p = \frac{1}{N} \sum_{i=1}^{N} (\psi(p_i))^{max}, \tag{2}$$

and analogous for $\langle \psi^{min} \rangle_p$;

5. maximum of the zonal-mean velocity potential $[\Phi]^{max}(p)$ at some predefined pressure level $p$, typically in the upper troposphere, e.g. at 200 hPa (Tanaka et al., 2004). The velocity potential is related to the wind divergence as $\nabla \cdot \mathbf{v} = \nabla^2 \Phi$;

6. minimum of the zonal-mean vertical velocity $[\omega]^{min}(p)$ at some predefined pressure level $p$, typically in the mid-troposphere, e.g. at 500 hPa (Wang, 2002), or a minimum $\omega$ within the tropical troposphere ($[\omega]^{min}$).

7. an average HC strength, which is obtained by spatially averaging the stream-function field in the latitude-pressure plane. For the northern HC, it yields

$$\psi_{NHC} = \langle \psi(\varphi, p) \rangle, \quad \text{for} \quad \psi(\varphi, p) > 0 \text{ and } (\varphi, p) \in [-20°, 40°] \times [50, 1000] \text{ hPa}, \tag{3}$$

where $\psi$ is uniformly sampled latitudinally, and vertically with a 50 hPa step. Wide latitudinal boundaries ensure that the Hadley cell is fully contained in every season (as shown in Fig. 1). An analogous measure $\psi_{SHC}$ is defined for the southern Hadley cell but with conditions $\psi < 0$ and meridional boundaries within $\varphi \in [-40°, 20°]$.

8. a normal-mode function based measure $I_M$, which is defined as the total energy of the zonal-mean unbalanced circulation. The index is obtained by projecting global geopotential and wind fields onto the normal-mode functions following Kasahara and Puri (1980); Žagar et al. (2015). The complex expansion coefficients $\chi_{k,n,m}$ associated with the inertia-gravity modes (IG) of the mean zonal state ($k = 0$) are then used to compute the total (kinetic plus potential) energy as

$$I_M = \sum_m g D_m \sum_n \chi_{k=0,n,m}^{\text{IG}} [\chi_{k=0,n,m}^{\text{IG}}]^* \tag{4}$$





where the indices $m$ and $n$ denote the vertical mode index and the meridional mode, respectively. For every $m$, $D_m$ denotes the associated eigenvalue known as the equivalent depth (e.g. Žagar et al., 2015, their Fig. 4), such that $D_1 >$
$D_2 > \ldots > D_M > 0$, where $M$ is the maximal vertical wave number. To define the HC strength, only coefficients corresponding to IG modes with $k = 0$, representing the zonal-mean state, are taken into account in Eq. (4). The radius of deformation on the equatorial $\beta-$plane, $a_e$, which defines the trapping scale of the modes, is defined by $D_m$: $a_e = \sqrt{\frac{gD_m}{\beta}}$, where $\beta = 2\Omega/R$, and $\Omega$ is Earth's angular velocity. Thus, larger $m$-s correspond to stronger equatorial trapping, e.g. for $D_7 = 708$ m the trapping scale is roughly $17°$. With all vertical and meridional modes included, the mean IG cir-
culation resides mainly within the tropics, and to a small extent near the major orographic features in the extratropics and in the polar winter stratosphere, as shown in Žagar et al. (2015, their Fig. 10) using ERA-Interim. For more details on the modal decomposition, see Appendix A. Figure A2 shows that the Hadley circulation is well represented by the zonal-mean unbalanced (IG) circulation (compare Fig. A2b vs. Fig. A2a).

The described indices have different properties. Indices (1)-(4) and (7) distinguish between the two Hadley cells, whereas
indices (5), (6) and (8) do not. Indices (1) to (3) do not capture the vertical inhomogeneities in the strength of the Hadley cell by definition. Index (4) captures the vertical but not the meridional inhomogeneity. Indices (5) and (6) describe the ascending branch of the Hadley circulation. New measure (7) by definition does not describe spatial inhomogeneities, but captures them by spatial averaging. The same applies to the new measure (8), which is by definition a global measure, but in large part explained by the Hadley circulation (Fig. A2) and also does not distinguish between the two HCs.
In the following section, we explore the sensitivity of the trends to different measures of the HC strength.

## 3    Sensitivity of the Hadley circulation trends to different measures

### 3.1    Comparison of the stream-function based measures

The sensitivity of the trends of the annual-mean and monthly-mean HC strength to the stream function-based measures (1)-(4) and (7), described in Section 2.3, is shown in Fig. 2 for ERA5 and in Fig. A3 for ERA-Interim. In both reanalyses, large
differences are observed between the trends of $\psi^{max}(p)$ at distinct pressure levels $p$ (measure (2)). In ERA5, the multiyear trend of the annual-mean NHC (leftmost column in Fig. 2a) is $0.7 \cdot 10^8$ kg s$^{-1}$ yr$^{-1}$ at 400 hPa and $2.3 \cdot 10^8$ kg s$^{-1}$ yr$^{-1}$ at 750 hPa. For the SHC (Fig. 2b), $\psi^{min}(p)$ strengthens by $0.9 \cdot 10^8$ kg s$^{-1}$ yr$^{-1}$ at 800 hPa and by $2.8 \cdot 10^8$ kg s$^{-1}$ yr$^{-1}$ at 400 hPa. In ERA-Interim, the NHC exhibits even differences in the sign of trends (leftmost major column in Fig. A3a); a strengthening trend of $2 \cdot 10^8$ kg s$^{-1}$ yr$^{-1}$ is present at 750 hPa and a weakening trend of $-0.4 \cdot 10^8$ kg s$^{-1}$ yr$^{-1}$ at 450 hPa. The SHC has an
insignificant trend of the annual-mean HC in the lower troposphere and a significant weakening of up to $-3 \cdot 10^8$ kg s$^{-1}$ yr$^{-1}$ in the upper troposphere (leftmost major column in Fig. A3b), i.e. opposite to what ERA5 shows.

The differences between trends of monthly-means at different pressure levels are even larger. For example, the February NHC exhibits a large and significant strengthening in the lower troposphere (700 hPa - 800 hPa) with trends around $7 \cdot 10^8$ kg s$^{-1}$ yr$^{-1}$ in both ERA5 (Fig. 2a) and ERA-Interim (Fig. A3a); however the trends in the mid-troposphere (400 hPa - 500 hPa)





are negative and mostly insignificant. Different magnitudes of the trends at distinct pressure levels can partly be explained by differences in the climatological-mean magnitude of the HC strength at different pressure levels. In general, the greater the mean HC magnitude, the greater the trend. The same feature can be observed in Fig. 1.

The differences in the trends of monthly-means at various pressure levels point at the unreliability of trend. Furthermore, magnitudes of the differences between indices are of the same order as the uncertainties of derived trends for individual indices.

Thus, by measuring the maximum HC strength at selected pressure level, e.g. 500 hPa (as in measure (2)), the estimated trends are affected by the limitation of the measure. At this level, the HC strength also exhibits a greater year-to-year variability of annual-mean and particularly monthly-mean variability (not shown), and consequently an increased uncertainty in the trend.

Another notable feature of Figs. 2b and A3b is a significant difference between the trends of the annual-mean SHC strength in ERA5 and ERA-Interim reanalyses; the SHC is strengthening in ERA5 but weakening in ERA-Interim. From July to October

the SHC is strengthening in both reanalyses, while from April to June it is weakening in ERA-Interim and strengthening in ERA5. The reasons for such discrepancies are likely in the data assimilation modelling and treatment of observations, and are therefore beyond the scope of this study.

Measure (1) exhibits significant year-to-year variability in the levels of $\psi^{max}$, observed also by Mitas and Clement, 2005. $\psi^{max}$ is switching between 350 hPa and 700 hPa levels in ERA5, and between 400 hPa and 650 hPa levels in ERA-Interim

(Fig. A5, magenta and red lines). In contrast, the level of $\psi^{min}$ remains roughly the same (700-750 hPa, blue and orange lines in Fig. A5) in both reanalyses throughout the studied period. Measure (1) also sometimes produces anomalous trends, which do not align with any of the other measures (e.g. in June and July NHC in ERA5, Fig. 2a).

Measure (2) does not capture the vertical inhomogeneity in the trend of the HC strength (as seen from $\psi$ at different levels in Figs. 2, A3). Measure (3) evaluates each Hadley cell in a spatially-fixed point throughout the observed period (1979-2018 in

our study). Thus, we expect it to be susceptible to potential meridional shifts of the mean Hadley circulation (Grise and Davis, 2020) or vertical shifts due to vertically expanding tropical troposphere (Hu and Vallis, 2019). As a single-point measure, it also suffers from spatial inhomogeneity of the trend of the HC strength, similar to measure (2). It can also produce spurious trends, such as the SHC trends in November in ERA5 (Fig. 2b, red bar), where the climatological maximum of the SHC is located at the Equator at 850 hPa pressure level (Fig. 1).

Vertically averaged maximum/minimum values of $\psi$ as in measure (4) reduce the discrepancies associated with the varying pressure levels of stream-function maxima and minima. Measure (4) also grasps the differences in the trends of the HC strength and averages them. Furthermore, such a measure averages out the differences between the trends at different pressure levels, as well as the uncertainty due to the choice of the pressure level in measure (2). However, Fig. 1 also revealed significant trend inhomogeneities in the meridional direction, e.g. between the ascending and the descending branches of the Hadley circulation,

which are addressed by the measure of average HC strength (i.e., by adding a meridional average).

The HC strength measured by (7) is on average weaker than in other $\psi$-based measures as spatial averaging leads to smaller magnitude of $\psi$ (not shown). Consequently, also the trends are smaller (Figs. 2, A3, rightmost violet bar in each major column). When trends are spatially more homogeneous, measure (7) exhibits relatively smaller uncertainties than the other measures (e.g. trends in monthly means of the NHC from March to May, ERA5, Fig. 1 and Fig. 2a), and conversely for spatially less





homogeneous trends (e.g. trends in monthly-means of the NHC from July to September and December, ERA5, Fig. 1 and
Fig. 2a). The average HC measure (7) thus provides an average over "extreme" local HC strength measures (1-4), as well as an
overall uncertainty. Note that Figs. 2, A3 merely showcase the stronger year-to-year variability of monthly means (compared
with year-to-year variability of annual means), as well as large discrepancies between $\psi$-measures at different levels (as also
seen from Figs. 1, A1), however from here on, we limit the analysis only to the trends of the annual-mean Hadley circulation.

### 3.2 Comparison of stream-function-based measures with other measures

The time-series of measures with different units and different mean magnitudes can be compared after their normalisation
which is in our case their respective climatological value for the 1979-2018 period, denoted $\langle\psi\rangle$. Results are shown in Fig. 3
for ERA5 reanalysis, including the normalized time-series of stream-function-based ($\psi$) measures (1)-(4) and (7), velocity-
potential ($\Phi$) based measures (5), pressure-velocity ($\omega$) based measures (6), and measure (8) describing the total energy of
the zonal-mean unbalanced circulation. Figure 4 and Table 1 present the trends of the normalized time-series, i.e. the relative
trends $(\partial\psi/\partial t)/\langle\psi\rangle$ in percentages per year, whereas Fig. 5 shows the correlations between the time-series of HC strength
derived from different measures.

In general, the normalized indices are well aligned in both HCs (Fig. 3, in grey colours), with a slightly larger spread over
a few periods (e.g. 1979-1982 in both HCs). The time-series of $\psi$-indices are better aligned for the SHC than the NHC, both
in ERA5 and ERA-Interim (Fig. A4). They are also highly correlated (Fig. 5), as expected from Fig. 3. For example, the time-
series derived from $\psi^{max}(p)$ (measure (2)) at neighbouring pressure levels (50 hPa apart) are highly correlated with correlation
coefficient $r > 0.98$, whereas $r > 0.94$ for measures 100 hPa apart. Absolute $\psi^{max}$ (measure (1)) correlates best with $\psi^{max}(p)$
at mid-tropospheric presssure levels (550-650 hPa), whereas absolute $\psi^{min}$ correlates best with $\psi^{min}(p)$ at lower-tropospheric
presssure levels (650-800 hPa). The result is in line with Fig. A5. Normalized measure (4) is highly correlated ($r > 0.9$) with
$\psi^{max}(p)$ and $\psi^{min}(p)$ at various levels.

A widely utilized HC strength measure $\psi^{max}(500\,\mathrm{hPa})$ also highly correlates ($r = 0.88$) with the average HC strength
measure (7), $\psi_{NHC}$. However, in the SHC, the stream function minimum at 500 hPa only moderately correlates ($r = 0.77$)
with the average SHC strength, $\psi_{SHC}$. On the other hand, $\psi^{min}$ at 700 hPa and 750 hPa has a high correlation with the
average HC strength ($r = 0.86$). These results suggest that the $\psi^{max}$-measures at pressure levels between 600 hPa and 500 hPa
are most representative of the overall changes in the NHC, whereas $\psi^{min}$ measures between 750 hPa and 700 hPa are most
representative for the SHC. The other single levels should probably be avoided as the HC strength indices.

Time-series of the other measures, i.e. $\psi^{max}$ or $\psi^{min}$, $\langle\psi^{max}\rangle_p$ or $\langle\psi^{min}\rangle_p$, $\psi(\varphi^{max}, p^{max})$ or $\psi(\varphi^{min}, p^{min})$ at the
location of the climatological maximum or minimum all highly correlate ($r = 0.82$ to $0.88$) with the average HC strength
measure as well. This means that the newly proposed average measure (7) is an adequate candidate for assessing the changes
of the HC strength.

Despite the high correlations, the relative trends of $\psi$-indices can differ significantly (Fig. 4), especially in ERA-Interim.
ERA5 (Fig. 4a,c) shows mostly significant strengthening from 0.09-0.36% yr$^{-1}$ for the NHC and 0.08-0.32% yr$^{-1}$ for SHC
(Table 1). In the NHC, the widely used measure $\psi^{max}$ (500 hPa) shows strengthening of 0.14% yr$^{-1}$ and is equal to the





**Table 1.** Annual-mean HC strength trends normalized by the climatological-mean values of the HC strength in ERA5 between 1979-2018. The trends derived from stream-function based measures (which distinguish between the NHC and the SHC), are separated by the horizontal black line from the trends, derived from other measures (which describe the two cells together). The values in the parentheses denote standard error of the trend estimates.

| NHC measure | trend (± unc.) [%/yr] | SHC measure | trend (± unc.) [%/yr] | HC measure | trend (± unc.) [%/yr] |
|---|---|---|---|---|---|
| $\psi^{max}$ | 0.177 (± 0.087) | $\psi^{min}$ | 0.136 (± 0.081) | $[\Phi]^{max}$ (150 hPa) | -0.375 (± 0.121) |
| $\psi^{max}$ (800 hPa) | 0.304 (± 0.104) | $\psi^{min}$ (800 hPa) | 0.082 (± 0.079) | $[\Phi]^{max}$ (200 hPa) | 0.110 (± 0.126) |
| $\psi^{max}$ (750 hPa) | 0.298 (± 0.108) | $\psi^{min}$ (750 hPa) | 0.125 (± 0.080) | $[\Phi]^{max}$ (250 hPa) | 1.136 (± 0.140) |
| $\psi^{max}$ (700 hPa) | 0.258 (± 0.108) | $\psi^{min}$ (700 hPa) | 0.160 (± 0.081) | $[\omega]^{min}$ (400 hPa) | 0.519 (± 0.170) |
| $\psi^{max}$ (650 hPa) | 0.213 (± 0.102) | $\psi^{min}$ (650 hPa) | 0.192 (± 0.082) | $[\omega]^{min}$ (500 hPa) | 0.349 (± 0.178) |
| $\psi^{max}$ (600 hPa) | 0.178 (± 0.091) | $\psi^{min}$ (600 hPa) | 0.214 (± 0.082) | $[\omega]^{min}$ (600 hPa) | 0.412 (± 0.187) |
| $\psi^{max}$ (550 hPa) | 0.157 (± 0.080) | $\psi^{min}$ (550 hPa) | 0.233 (± 0.081) | $[\omega]^{min}$ | 0.738 (± 0.155) |
| $\psi^{max}$ (500 hPa) | 0.140 (± 0.073) | $\psi^{min}$ (500 hPa) | 0.257 (± 0.079) | $I_M$ | 0.072 (± 0.074) |
| $\psi^{max}$ (450 hPa) | 0.116 (± 0.070) | $\psi^{min}$ (450 hPa) | 0.280 (± 0.077) | | |
| $\psi^{max}$ (400 hPa) | 0.094 (± 0.070) | $\psi^{min}$ (400 hPa) | 0.319 (± 0.075) | | |
| $\psi(\varphi^{max}, p^{max})$ | 0.294 (± 0.106) | $\psi(\varphi^{min}, p^{min})$ | 0.177 (± 0.079) | | |
| $\langle\psi^{max}\rangle_p$ | 0.136 (± 0.077) | $\langle\psi^{min}\rangle_p$ | 0.183 (± 0.073) | | |
| $\psi^{NHC}$ | 0.358 (± 0.071) | $\psi_{SHC}$ | 0.223 (± 0.070) | | |

trend of $\langle\psi^{max}\rangle_p$, while $\psi^{max}$ increases by 0.18% yr$^{-1}$. $\psi(\varphi^{max}, p^{max})$ and $\psi_{NHC}$ show larger trends with strengthening of

0.29% yr$^{-1}$ and 0.36% yr$^{-1}$, respectively. The two measures which perform spatial averaging, $\langle\psi^{max}\rangle_p$ and $\psi_{SHC}$, suggest strengthening of the southern cell by 0.18% yr$^{-1}$ and 0.22% yr$^{-1}$, respectively. The relative trends derived from the average HC strength measure (7) show mildly reduced uncertainty compared to the other stream-function-based point measures, in line with the results of Section 3.1.

The time-series derived from $\omega$-indices have much higher temporal oscillations compared with $\psi$-indices (Fig. 3), however

the maxima and minima are fairly aligned with $\psi$-indices, though with larger anomalies, which is captured also by their moderate correlations ($r = 0.3$ to 0.5 for the SHC and 0.4 to 0.65 for the NHC) (Fig. 5, A6). However, the average HC strength (measure (7)), $\psi_{NHC}$ and $\psi_{SHC}$, correlates better with the $\omega$-indices: $r = 0.67$ to 0.80 for the NHC and 0.63 to 0.71 for the SHC. The measure (7) also correlates better with the $\Phi$-indices than other $\psi$-indices, particularly with $[\Phi]^{max}$ at 200 hPa and 250 hPa. This further implies that the average HC strength (measure (7)) captures also the changes in the HC in regions of

ascending motion. The correlation of $[\Phi]^{max}$ at 150 hPa with other measures is low and mostly insignificant, suggesting that the 150 hPa level might already be in the tropical tropopause.

The velocity-potential-based measures $[\Phi]^{max}(p)$ show much larger magnitude of the trends compared with the other measures. They are also very susceptible to the applied pressure level, a similar issue as for the $\psi$-indices. Therefore, this measure is also likely susceptible to the potential future changes in the depth of the tropical troposphere. For example, $[\Phi]^{max}$ (250





hPa) in ERA5 shows a strengthening trend of 1.14% yr$^{-1}$, at 200 hPa roughly 0.11% yr$^{-1}$, whereas $[\Phi]^{max}$ at 150 hPa shows a weakening trend of -0.38% yr$^{-1}$ (Fig. 4a), an outlier among the other measures. The differences among trend magnitudes are even larger in ERA-Interim (Fig. 4b).

The trends derived from the $\omega$-indices align reasonably well with the trends derived from the $\psi$-indices. In particular, $[\omega]^{min}$ at 500 hPa (dark grey bar in Fig. 4) shows good agreement with the average HC strength (measure (7)), but with more than

twice as large uncertainty due to larger variability of the $\omega$-indices, as revealed in Fig. 3. As for the other point measures, the derived trends of the $\omega$-based HC strength are strongly susceptible to the choice of a pressure level (this is again more pronounced in the ERA-Interim).

The total energy of the zonal-mean unbalanced circulation $I_M$ has strengthened in the 1979-2018 period in both ERA5 and ERA-Interim with a rate of 0.07% yr$^{-1}$ and 0.28% yr$^{-1}$ (Fig. 4). The uncertainty of the trend is relatively small compared

to the other measures (Tables 1, A1). The sign of the derived trends in ERA5 is consistent with other measures in ERA5, although the magnitude is smaller. However, $I_M$ suggests strengthening of the global unbalanced circulation also in ERA-Interim (Fig. 4b,d), a trend opposite to that derived from the stream-function-based indices. Furthermore, the correlation of the unbalanced energy index with other indices is low and insignificant (Fig. 5). Insignificant correlations are not surprising as this index is largely different from all other indices. First, it is the total energy measure; the kinetic energy part is due to

both components of the horizontal flow and a contribution to the energy comes also from outside the tropics and from the stratosphere. We argue that the part of $I_M$ from the extra-tropics and the stratosphere is unimportant for the overall signal, but it may be important for the trends. Zagar et al. (2020) discussed unbalanced circulation in ERA5 and ERA-Interim in relation to the data assimilation behind the two reanalyses. They showed that in spite of the differences, the two datasets agree on the positive trend in the most energetic large-scale features of tropical circulation.

To quantify the role of the stratospheric circulation to the uncertainties in the trends in $I_M$, we compared $I_M$ in ERA-Interim focusing on levels up to 100 hPa only. It revealed a smaller trend though still positive and a somewhat higher correlation with the other indices (not shown).

Given the importance of the mixed Rossby-gravity (MRG) waves in the Hadley circulation (Hoskins et al., 2020), we also tested an extension of $I_M$, which consists of adding the MRG wave energy to the zonal-mean unbalanced energy (4). In

this case, the relative trend increased by a slight margin, while the correlation with other measures remained insignificant (not shown). Furthermore, performing the summation (4) for a subset of vertical modes (e.g. $m \geq 9$), thereby reducing the stratospheric and high-latitude contributions to the $I_M$, results both in greater correlations with other measures, and in a larger relative trend, which is better aligned with other measures (not shown).

## 4 Conclusions

In this study, we analysed a number of indices of the Hadley circulation strength including indices based on the mass-weighted mean meridional stream-function, velocity potential, pressure velocity $\omega$, and the total energy of the zonal-mean unbalanced circulation. The indices were applied to ERA5 and ERA-Interim reanalysis data between 1979 and 2018. While ERA5 is our





main dataset, its comparison with ERA-Interim provides confidence that the observed characteristics of a particular measure are not an isolated feature of ERA5 reanalysis. However, the comparison is not straightfoward as the two reanalyses differ in

their representation of the unbalanced tropical circulation. This was made evident by a new HC strength measure defined as the global total energy of the unbalanced zonal-mean circulation. Another newly proposed measure describes the average strength of the NHC and SHC using the average stream function and is therefore insensitive to spatial inhomogeneities.

By analysing the temporal changes of the stream function changes in the latitude-pressure plane, we showed that the HC strength trends are spatially inhomogeneous, both meridionally and vertically (Figs. 1,A1), particularly in ERA-Interim. Dis-

tinct HC strength measures resulted in significantly different and sometimes even opposing trends, decreasing our prospects to draw firm conclusions on the circulation changes. The two new measures of the HC strength are characterized by a smaller uncertainty of the derived trends compared to the current measures of the HC strength, likely due to spatial averaging (average stream function) or the integration (energy of zonal-mean unbalanced circulation). However, the normal modes based index is affected by its global definition meaning that the unbalanced zonal-mean circulation outside the tropical and subtropical

troposphere is also accounted for. Future work can refine the index.

In light of all the results, we recommend using the measure of the average HC strength (measure (7) in Section 2.3) whenever interested in the variability and trends of the HC strength. Having said this, usage of new and established measures will ultimately depend on the purpose of a study.

Presented opposing trends suggest that the contribution of physical mechanisms that drive the Hadley cells and govern their

strength (e.g. diabatic heating, friction, eddy heat and momentum fluxes, static stability, etc.) are likely to vary with the chosen HC strength measure (Chemke and Polvani, 2019; Zaplotnik et al., 2021). For example, the friction should affect the HC strength trends more if the measure $\psi^{max}(p)$ is taken at some lower-tropospheric pressure level, whereas its impact is likely reduced when $\psi^{max}(p)$ is evaluated at mid-to-upper tropospheric levels. However, a detailed analysis of these effects is beyond the scope of this study and will be pursued in the future.

Our results confirm that caution is needed when comparing HC trends from different studies using different measures of the HC strength. A unified index of the Hadley circulation would allow a better estimation of the likelihood of the future changes in the global atmospheric circulation (e.g., Stocker et al., 2013).

*Code and data availability.*   Scripts are available upon request. The ERA-Interim and ERA5 reanalysis datasets are available from http://www.ecmwf.int. The data were obtained using Copernicus Climate Change Service information 2021. Data used to generate Figs. 2 to 5 and Figs. A3 to A6 are

publicly available at https://github.com/zaplotnik/Hadley-cell-strength and publsihed in Zenodo data repository: https://zenodo.org/record/5135222?.YPzCMXUzb6c.

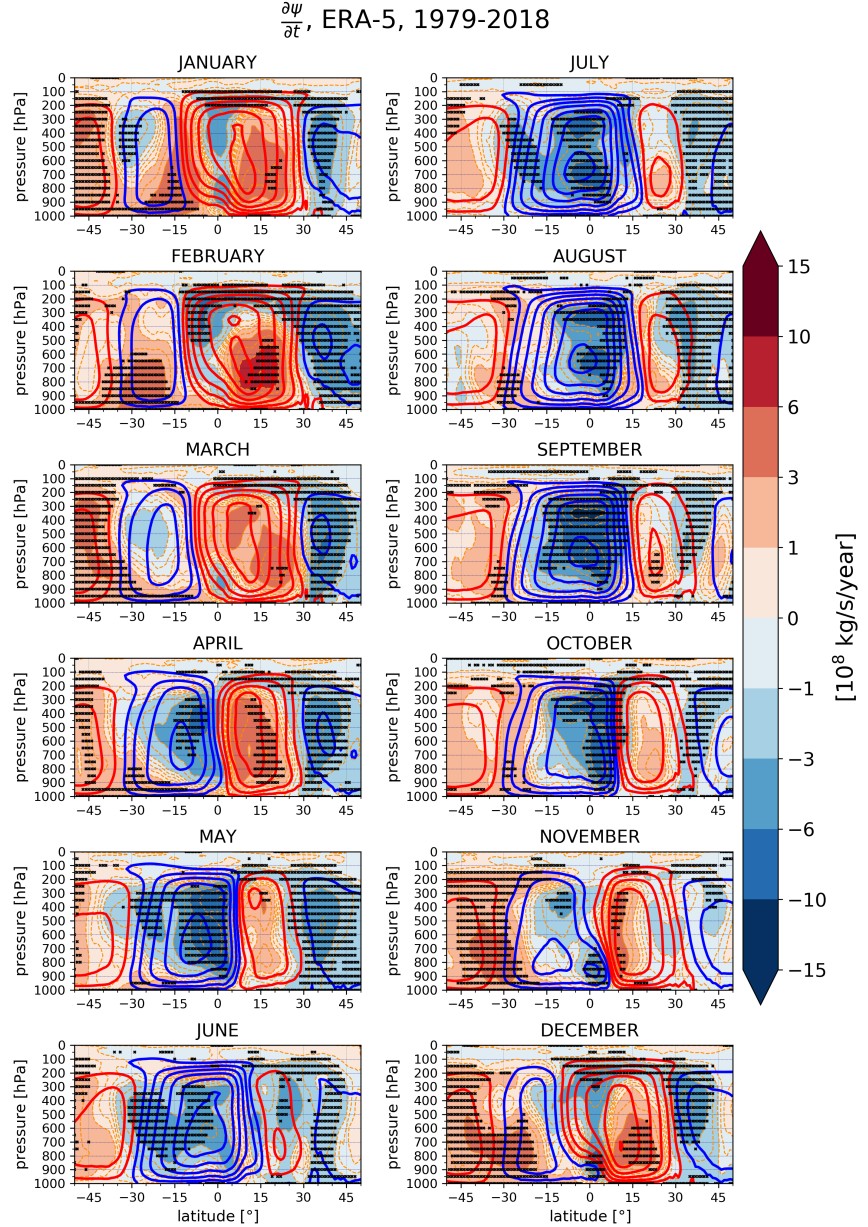

**Figure 1.** Monthly-mean climatology of the Hadley Circulation (red and blue contours) and its trends (shading) in ERA5 reanalysis between 1979 - 2018. Red contours indicate positive climatological stream function values, i.e. $(0.1, 0.3, 0.6, 1, 1.5, 2, 2.5) \cdot 10^{11}$ kg s$^{-1}$ and blue contours their negative equivalents, i.e. $(-0.1, -0.3, -0.6, -1, -1.5, -2, -2.5) \cdot 10^{11}$ kg s$^{-1}$. Crosses indicate the statistically significant trends at the 95% confidence level. Note that the overlapping of contours and shading of the same colour indicates strengthening of the cell, while overlapping of different colours indicates cell weakening.





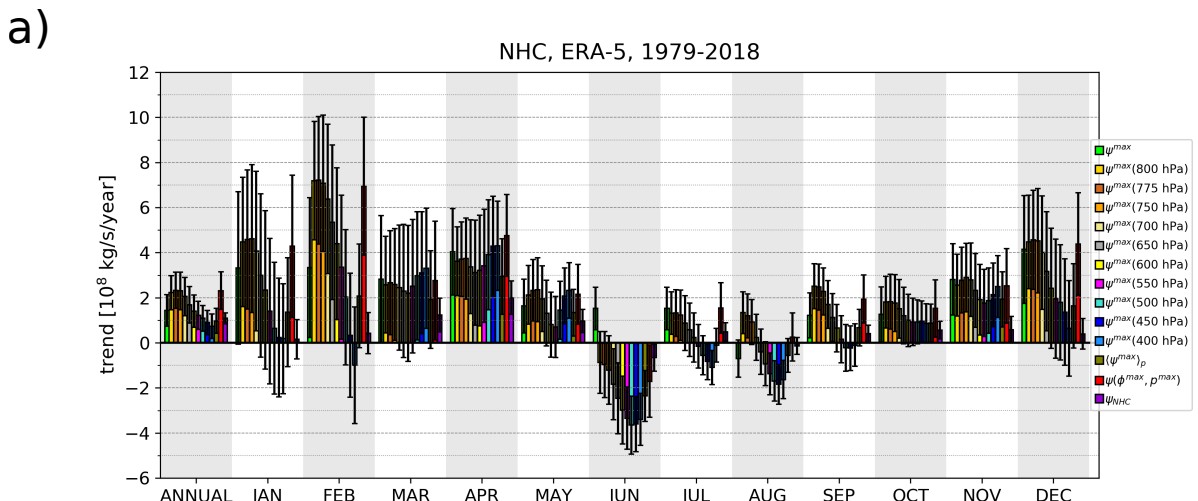

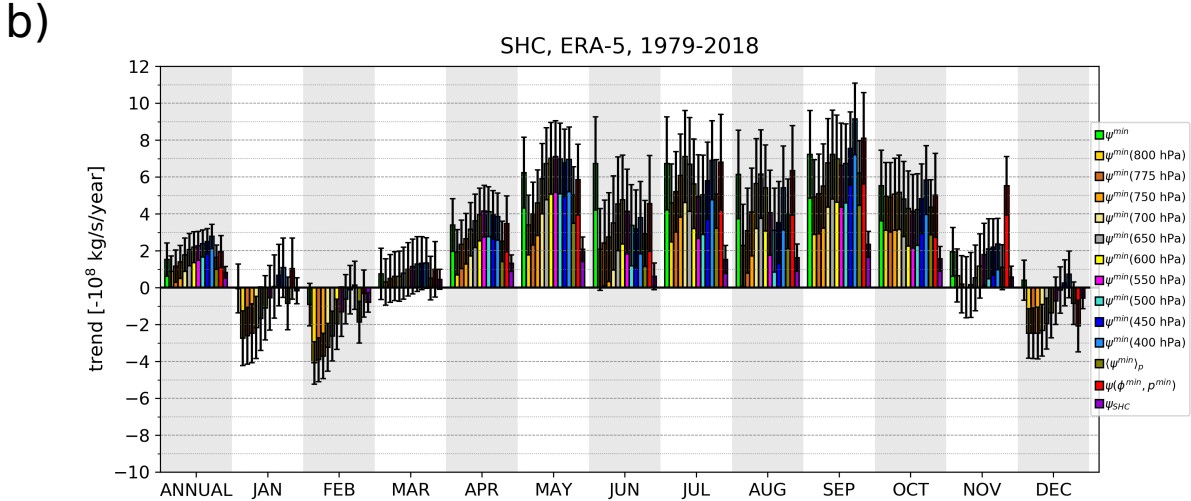

**Figure 2.** Trends of the NHC strength (a) and SHC strength (b) in ERA5 reanalysis between 1979-2018 for different stream-function-based measures from Section 2.3. Annual-mean trends of the HC strength are shown in the first column, while monthly-mean trends are shown in the other columns (as labeled). Different measures of the HC strength are shown in the legend, e.g. in (a) for the NHC: $\psi^{max}$ denotes annual/monthly stream function maximum (measure (1)), $\psi^{max}$ at 400 hPa, 450 hPa, etc. denotes annual/monthly stream function maximum at respective pressure level (measure (2)), and $\psi(\varphi^{max}, p^{max})$ denotes that the trends are measured at the point of the maximum stream function in a multiyear average of the NHC strength (measure (3)). $\langle \psi^{max} \rangle_p$ denotes vertically averaged $\psi^{max}$ between 200 hPa and 900 hPa (measure (4), Eq. 2) and $\psi_{NHC}$ denotes measure of average HC strength (measure (7), Eq. 3). Analogous notations are used for the stream function minimum for the SHC in (b). Note that values in (b) are multiplied by (-1), thus positive values in both (a) and (b) indicate strengthening of the cell. Black error bars indicate standard error of the trend estimates.





a)

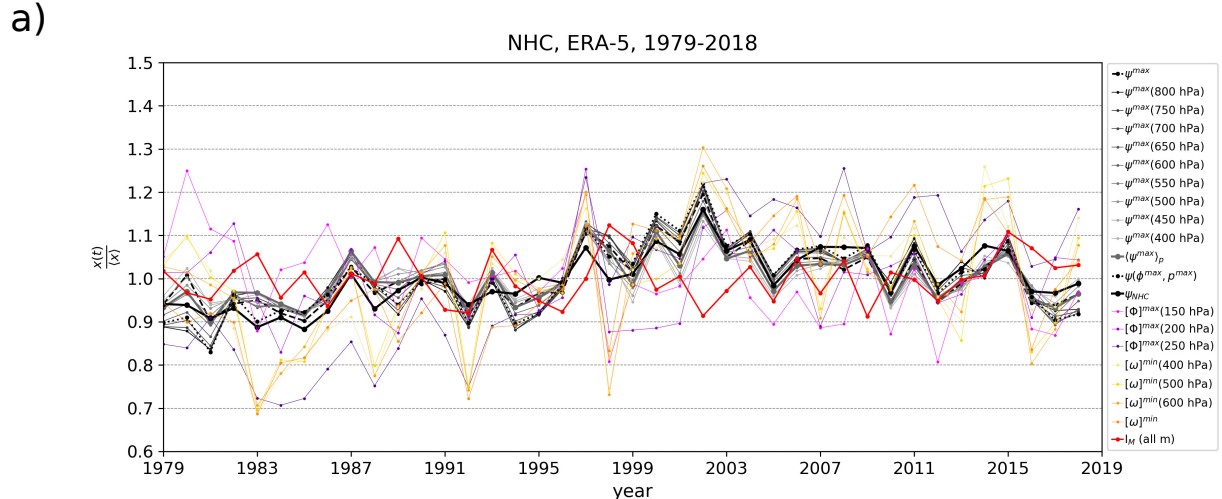

b)

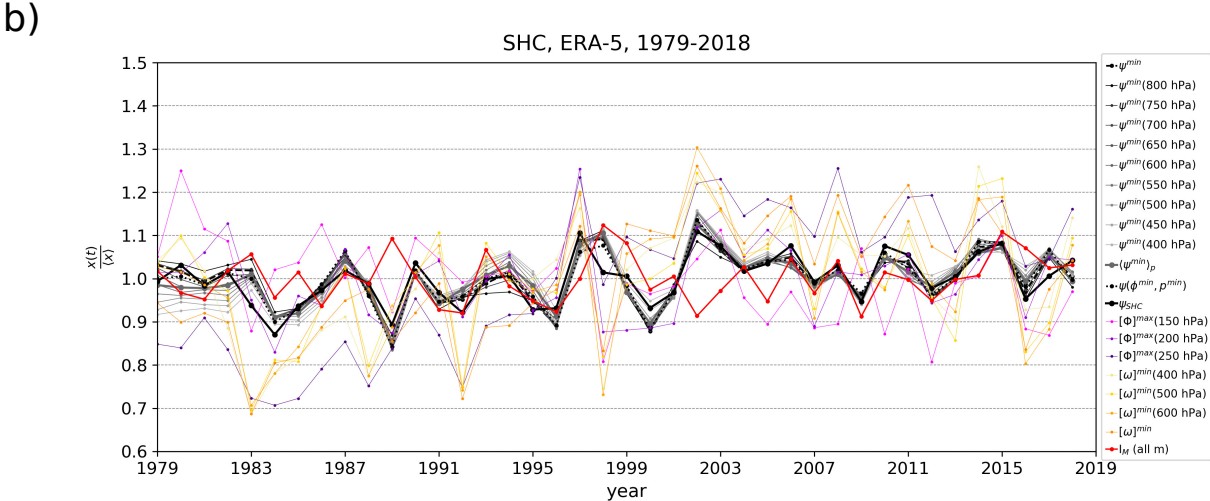

**Figure 3.** Time-series of the NHC strength (a) and the SHC strength (b) in ERA5 reanalysis between 1979-2018 for different measures from Section 2.3. Time-series are normalized by their 1979-2018 climatological mean. Different stream-function-based measures of the HC strength are shown in the legend (in grey colours), e.g. in (a) for the NHC: $\psi^{max}$ denotes the annual/monthly stream function maximum (measure (1)); $\psi^{max}$ at 800 hPa, 750 hPa, 700 hPa, etc. denotes the annual/monthly stream function maximum at respective pressure level (measure (2)); $\psi(\varphi^{max}, p^{max})$ denotes that the HC strength is measured at the point of the maximum stream function in a multiyear average of the NHC strength (measure (3)); $\langle\psi^{max}\rangle_p$ denotes the vertically averaged $\psi^{max}$ between 200 hPa and 900 hPa (measure (4), Eq. 2); and $\psi_{NHC}$ denotes the measure of the average HC strength (measure (7), Eq. 3). Analogous notations are used for the stream function minimum for the SHC in (b). The following measures do not distinguish between the two Hadley cells, but describe the Hadley circulation as a whole (their time-series are thus the same for NHC in (a) and SHC in (b)): $[\Phi]^{max}(p)$ denotes the maximum of the zonal-mean velocity potential at different pressure levels (measure (5), orange colours); $[\omega]^{min}(p)$ denotes the minimum of the zonal-mean vertical velocity (measure (6), violet colours); and $I_M$ denotes the normal-modes-based index of the Hadley circulation (measure (8), red colour).



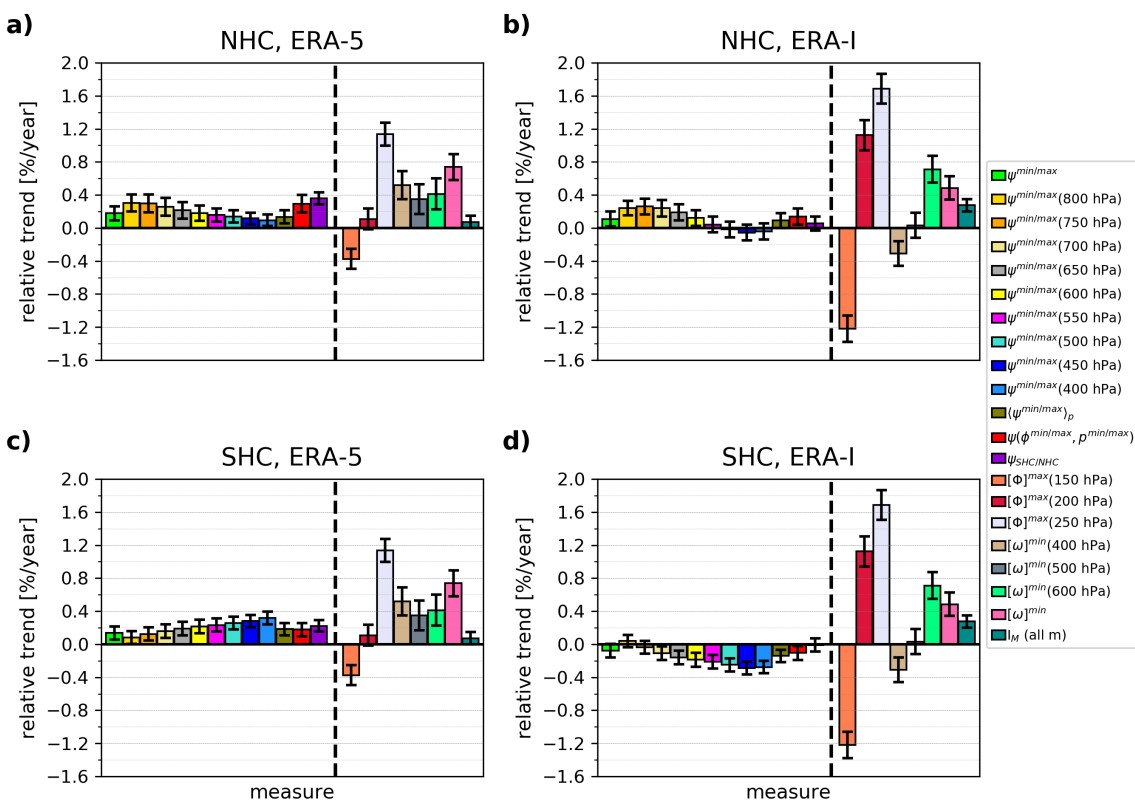

**Figure 4.** Annual-mean HC strength trends normalized by the climatological-mean values of the HC strength in the NH (a,b) and SH (c,d) in ERA5 (a,c) and ERA-Interim (b,d) reanalyses between 1979-2018 for different measures of HC strength defined in Section 2.3. Different measures of the HC strength are shown in the legend, e.g. for the NHC: $\psi^{max}$ at 400 hPa, 450 hPa,... denotes the annual-mean stream function maximum at respective pressure level (measure (2)); $\psi^{max}$ denotes the annual-mean stream function maximum (measure (1)); $\psi(\varphi^{max}, p^{max})$ denotes that the trends are measured at the point of the maximum stream function in a multiyear average of the NHC strength (measure (3)); $\langle \psi^{max}(t) \rangle_p$ denotes the vertically averaged $\psi^{max}(t)$ between 200 hPa and 900 hPa (Eq. 2) (measure (4)); and $\psi_{NHC}$ denotes the measure of average HC strength (Eq. 3) (measure (7)). Analogous notations are used for the stream function minimum for the SHC. The following measures do not distinguish between the two Hadley cells, but describe the Hadley circulation as a whole (their results are the same for the NHC and the SHC, and are separated by the vertical black dashed line): $[\Phi]^{max}$ denotes the maximum of the zonal-mean velocity potential (measure (5)), $[\omega]^{min}$ denotes the minimum of the zonal-mean vertical velocity (measure (6)), and $I_M$ denotes the normal-modes based index of the Hadley circulation (measure (8)). Note that positive values in all panels indicate strengthening of the NHC and SHC. Black error bars indicate standard error of the trend estimates.



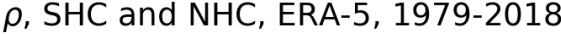

Figure 5. Correlations of time-series, derived from different measures of the Hadley cell strength, described in Section 2.3, for 1979-2018 period in ERA5 reanalysis. The correlations for the northern Hadley cell are shown in the upper right part of the matrix, whereas the southern Hadley cell correlations are represented in the lower-left part. Time-series of $[\omega(p)]^{min}$ are multiplied by $(-1)$ so that more positive values correspond to HC strengthening. Similarly, the time-series of the stream-function based measures $\psi_{min}$ are also multiplied by $(-1)$ so that more positive values correspond to HC strengthening. Only correlations exceeding 95% significance threshold are shown.



**Appendix A**

MODES software (Žagar et al., 2015) is used to perform scale- and circulation-type-dependent decomposition of the 3D dynamical fields: the zonal wind $u$, meridional wind $v$ and modified geopotential $h = P/g$ with $P = \Phi + RT_0 \ln p_s$. Here, $\Phi$ represents the geopotential, $R$ is the gas constant, $T_0(p)$ is globally-averaged temperature on a certain pressure level. The input data vector $[u, v, h]^T$ is decomposed using separable series of $M$ orthogonal vertical structure functions $G_m(p)$ and series of horizontal structure functions (Hough harmonics) $\mathbf{H}_n^k(\lambda, \varphi; m)$, which consist of $2K + 1$ zonal waves and $R$ meridional waves:

$$
\begin{bmatrix} u(\lambda, \phi, p) \\ v(\lambda, \phi, p) \\ h(\lambda, \phi, p) \end{bmatrix} = \sum_{m=1}^{M} G_m(p) \, \mathbf{S}_m \sum_{n=1}^{R} \sum_{k=-K}^{K} \chi_{knm} \underbrace{\Theta_n^k(\varphi; m) \, e^{ik\lambda}}_{\mathbf{H}_n^k(\lambda, \varphi; m)},
\tag{A1}
$$

where $\mathbf{S}_m = \mathrm{diag}(\sqrt{gD_m}, \sqrt{gD_m}, D_m)$ is a diagonal matrix, $g$ is gravitatonal acceleration. $D_m$ is an equivalent depth of the vertical mode $m$ and couples the vertical and horizontal structure functions. $\chi_{knm}$ are the spectral Hough coefficients. $\Theta_n^k(\varphi; m)$ is meridional vector function consisting of multivariately related components $[U_n^k, -iV_n^k, Z_n^k]^T(\varphi; m)$. For every vertical mode $m$, the system of horizontal structure equations applies

$$
\frac{\partial u}{\partial t} - 2\Omega v \sin \varphi + \frac{g}{R \cos \varphi} \frac{\partial h}{\partial \lambda} = 0
$$
$$
\frac{\partial v}{\partial t} + 2\Omega u \sin \varphi + \frac{g}{R} \frac{\partial h}{\partial \varphi} = 0
\tag{A2}
$$
$$
\frac{\partial h}{\partial t} + D_m \nabla \cdot \mathbf{v} = 0.
$$

The equations can be made dimensionless by taking $\widetilde{u} = u'/\sqrt{gD_m}$, $\widetilde{v} = v'/\sqrt{gD_m}$, $\widetilde{h} = h'/D_m$ and $\widetilde{t} = 2\Omega t$, so that

$$
\frac{\partial}{\partial \widetilde{t}} \mathbf{W}_m + \mathbf{L} \mathbf{W}_m = \mathbf{0},
\tag{A3}
$$

where $\mathbf{W}_m = [\widetilde{u}, \widetilde{v}, \widetilde{h}]^T$ and $\mathbf{L}$ is the linear differential matrix operator

$$
\mathbf{L} = \begin{bmatrix} 0 & -\sin \varphi & \frac{\gamma}{\cos \varphi} \frac{\partial}{\partial \lambda} \\ \sin \varphi & 0 & \gamma \frac{\partial}{\partial \varphi} \\ \frac{\gamma}{\cos \varphi} \frac{\partial}{\partial \lambda} & \frac{\gamma}{\cos \varphi} \frac{\partial}{\partial \varphi}(\cos \varphi(\cdot)) & 0 \end{bmatrix}.
\tag{A4}
$$

$\gamma$ is a dimensionless parameter defined as the ratio of shallow-water gravity wave speed and twice the rotation speed of Earth, $\gamma = \sqrt{gD_m}/(2R\Omega)$. The third equation in system (A2) now becomes

$$
\frac{\partial}{\partial \widetilde{t}} \widetilde{h}_m + \frac{\sqrt{gD_m}}{2\Omega} (\nabla \cdot \widetilde{\mathbf{v}}_m) = 0,
\tag{A5}
$$

The solution ansatz can be expressed by assuming separability of time-dependent and space-dependent solutions, i.e.

$$
\mathbf{W}_m(\lambda, \phi, \widetilde{t}) = \mathbf{H}_n^k(\lambda, \phi; m) e^{-i\widetilde{\sigma}_{knm} \widetilde{t}},
\tag{A6}
$$

where $\widetilde{\sigma}_{knm}$ is dimensionless frequency, and $\mathbf{H}_n^k(\lambda, \varphi; m)$ are the associated horizontal structure functions, which are used in the expansion (A1).



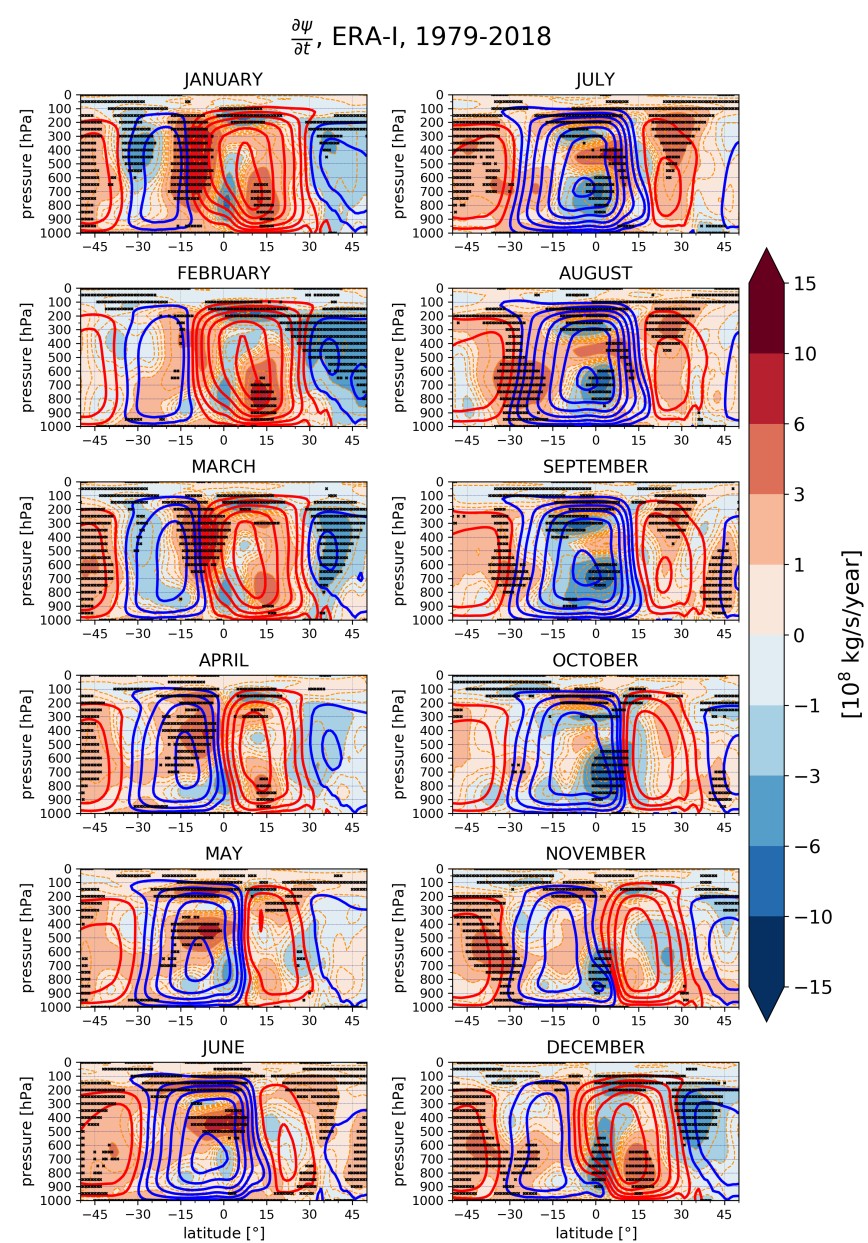

**Figure A1.** As in Fig. 1, but for the ERA-Interim reanalysis.

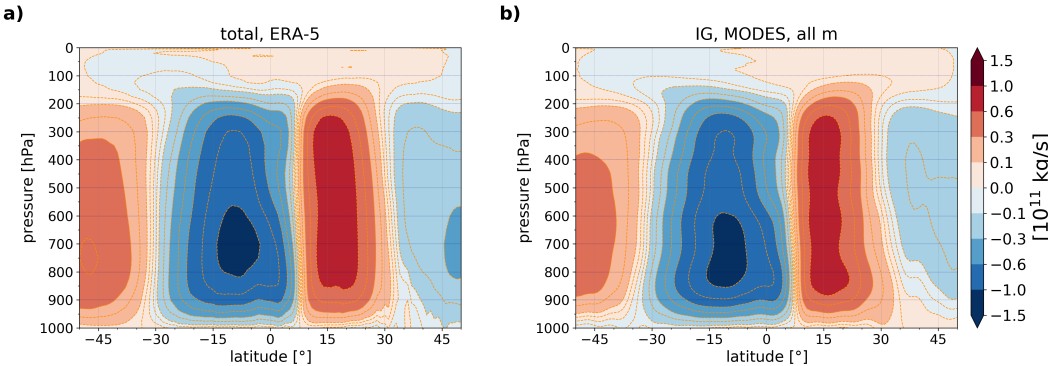

**Figure A2.** The 2018 mean Hadley Circulation (red and blue contours) in ERA5 reanalysis computed from (a) total fields of zonal-mean meridional wind and (b) unbalanced (inertia-gravity) fields. Contours indicate values of stream function $\psi$.





a)

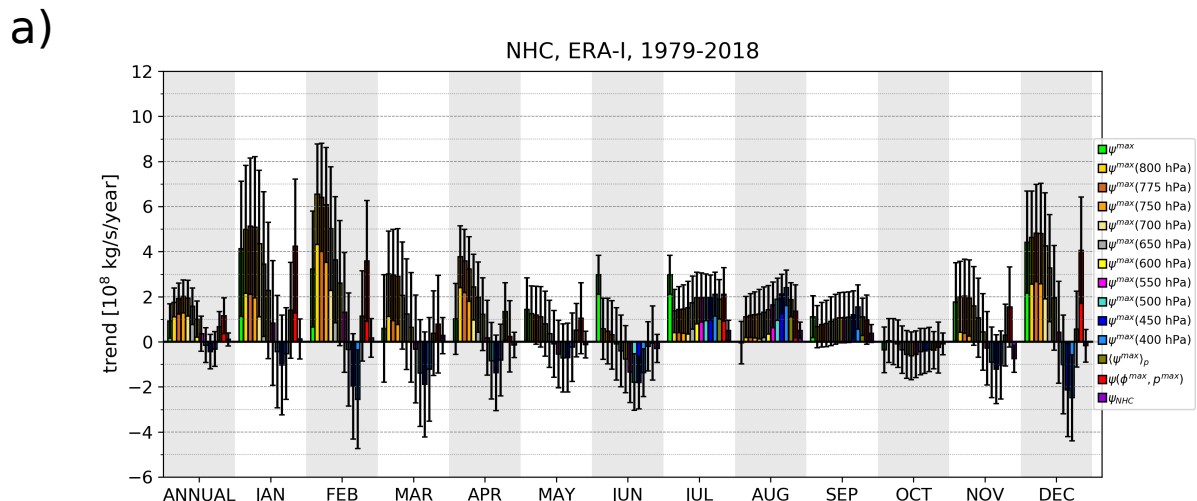

b)

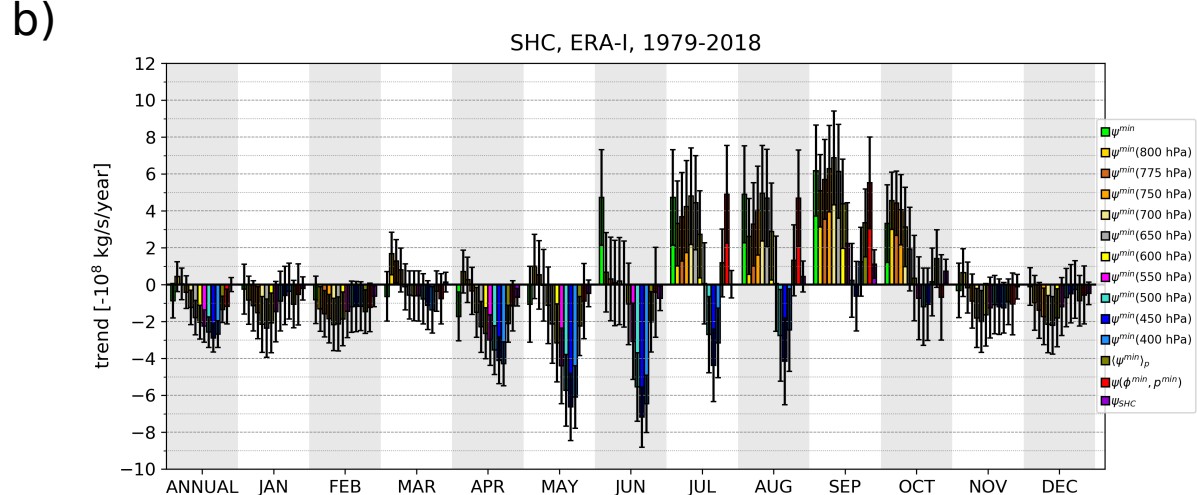

**Figure A3.** As in Fig. 2, but for the ERA-Interim reanalysis.




a)

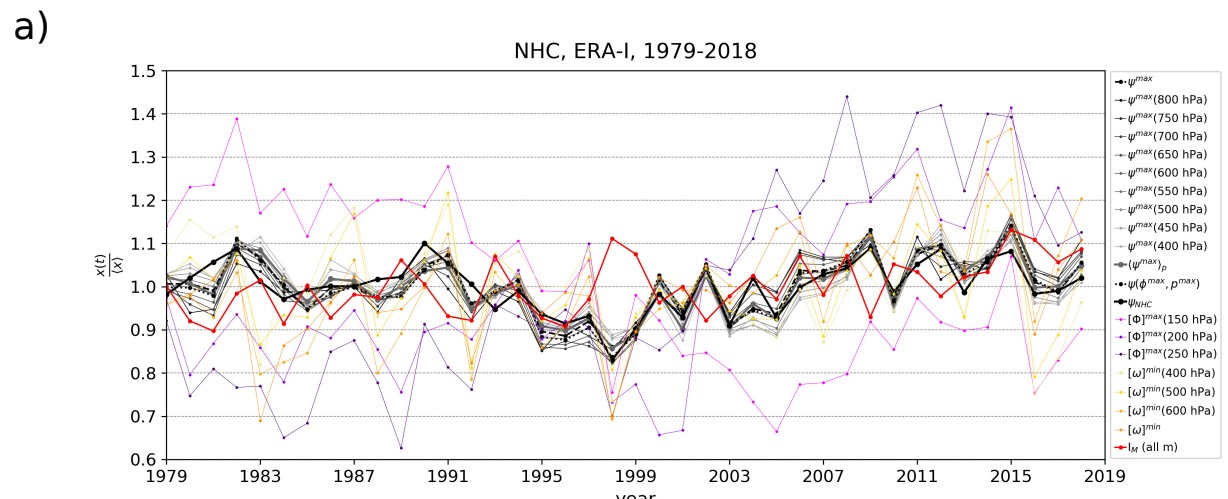

b)

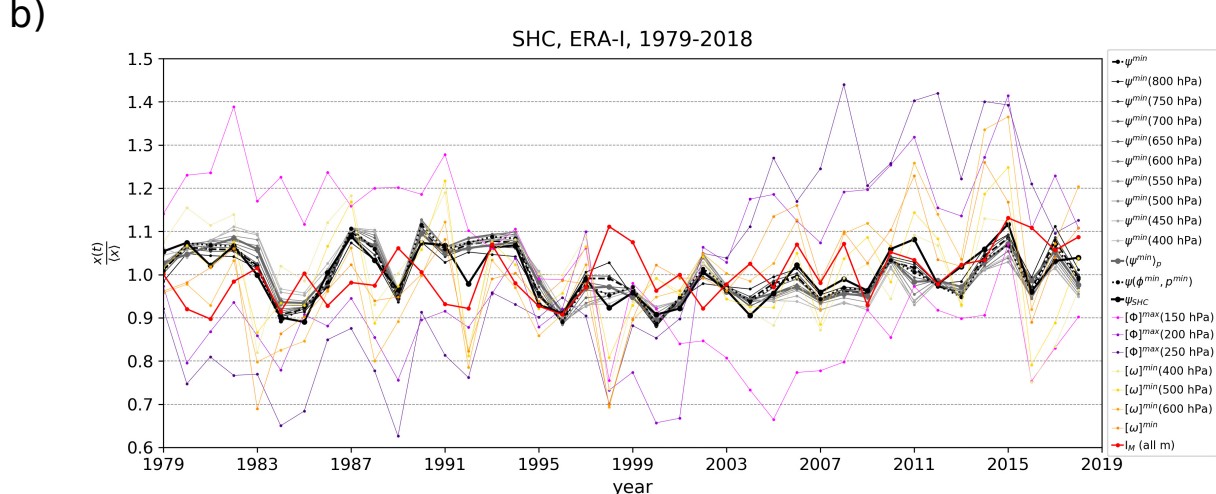

**Figure A4.** As in Fig. 3, but for the ERA-Interim reanalysis.



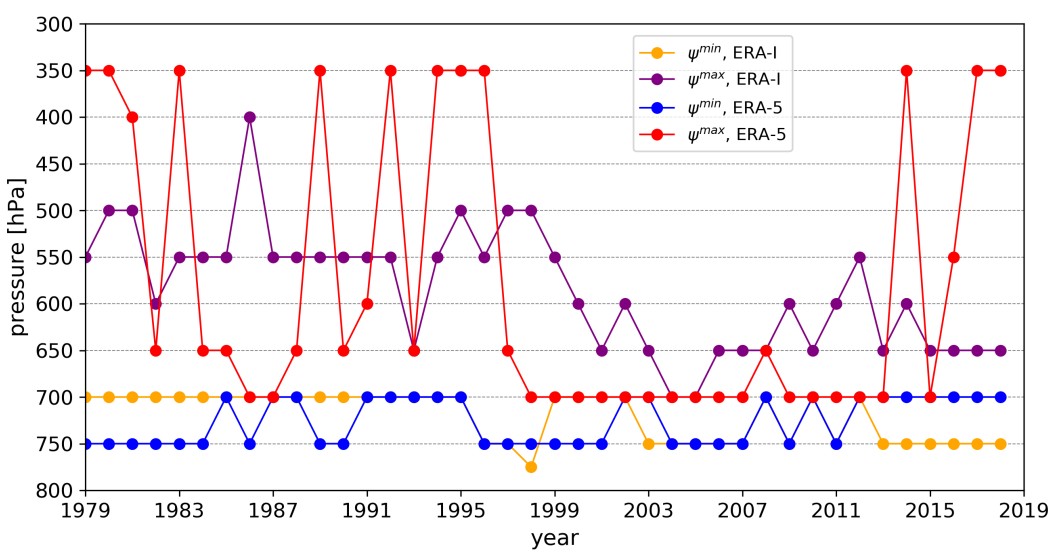

**Figure A5.** Level of maximum/minimum stream function in annual-mean Hadley circulation between 1979-2018 in ERA5 and ERA-Interim reanalyses.





**Figure A6.** As in Fig. A6, but for ERA-Interim.





**Table A1.** As in Table A1, but for ERA-Interim.

| NHC measure | trend ($\pm$ unc.) [%/yr] | SHC measure | trend ($\pm$ unc.) [%/yr] | HC measure | trend ($\pm$ unc.) [%/yr] |
|---|---|---|---|---|---|
| $\psi^{max}$ | 0.109 ($\pm$ 0.091) | $\psi^{min}$ | -0.077 ($\pm$ 0.082) | $[\Phi]^{max}$ (150 hPa) | -1.221 ($\pm$ 0.160) |
| $\psi^{max}$ (800 hPa) | 0.240 ($\pm$ 0.088) | $\psi^{min}$ (800 hPa) | 0.039 ($\pm$ 0.075) | $[\Phi]^{max}$ (200 hPa) | 1.125 ($\pm$ 0.183) |
| $\psi^{max}$ (750 hPa) | 0.259 ($\pm$ 0.094) | $\psi^{min}$ (750 hPa) | -0.036 ($\pm$ 0.079) | $[\Phi]^{max}$ (250 hPa) | 1.686 ($\pm$ 0.181) |
| $\psi^{max}$ (700 hPa) | 0.239 ($\pm$ 0.098) | $\psi^{min}$ (700 hPa) | -0.109 ($\pm$ 0.082) | $[\omega]^{min}$ (400 hPa) | -0.310 ($\pm$ 0.148) |
| $\psi^{max}$ (650 hPa) | 0.191 ($\pm$ 0.098) | $\psi^{min}$ (650 hPa) | -0.161 ($\pm$ 0.084) | $[\omega]^{min}$ (500 hPa) | 0.032 ($\pm$ 0.151) |
| $\psi^{max}$ (600 hPa) | 0.121 ($\pm$ 0.096) | $\psi^{min}$ (600 hPa) | -0.188 ($\pm$ 0.084) | $[\omega]^{min}$ (600 hPa) | 0.710 ($\pm$ 0.162) |
| $\psi^{max}$ (550 hPa) | 0.043 ($\pm$ 0.094) | $\psi^{min}$ (550 hPa) | -0.212 ($\pm$ 0.082) | $[\omega]^{min}$ | 0.484 ($\pm$ 0.141) |
| $\psi^{max}$ (500 hPa) | -0.018 ($\pm$ 0.093) | $\psi^{min}$ (500 hPa) | -0.248 ($\pm$ 0.080) | $I_M$ | 0.276 ($\pm$ 0.075) |
| $\psi^{max}$ (450 hPa) | -0.055 ($\pm$ 0.095) | $\psi^{min}$ (450 hPa) | -0.287 ($\pm$ 0.077) | | |
| $\psi^{max}$ (400 hPa) | -0.041 ($\pm$ 0.098) | $\psi^{min}$ (400 hPa) | -0.278 ($\pm$ 0.075) | | |
| $\psi(\varphi^{max},p^{max})$ | 0.138 ($\pm$ 0.096) | $\psi(\varphi^{min},p^{min})$ | -0.105 ($\pm$ 0.084) | | |
| $\langle\psi^{max}\rangle_p$ | 0.092 ($\pm$ 0.090) | $\langle\psi^{min}\rangle_p$ | -0.141 ($\pm$ 0.074) | | |
| $\psi^{NHC}$ | 0.054 ($\pm$ 0.086) | $\psi_{SHC}$ | -0.009 ($\pm$ 0.079) | | |

*Author contributions.* MP performed numerical analysis and generated all figures. ŽZ devised the research, performed the modal analysis and wrote the first draft of the manuscript. NŽ oversaw modal analysis. LB and NŽ provided additional insight and helped improve the manuscript for the final version.

*Competing interests.* The authors declare that they have no conflict of interest.

*Acknowledgements.* MP is supported by ARRS Programme P1-0188. ŽZ is funded by the Slovenian Research Agency project J1-9431 and is also supported by ARRS Programme P1-0188. LB is supported by Trond Mohn Foundation. Research of N. Ž. contributes to the Cluster of Excellence 405 "CLICCS-Climate, Climatic Change, and Society" of the Center for Earth System Research and Sustainability (CEN) of Universität Hamburg.



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
