# Peer review of "Metrics of the Hadley circulation strength and associated circulation trends"

_Weather and Climate Dynamics, 2021_

## Author Comment (AC2)

**General comments:**

**Comment:** It's nice to see Pikovnik et al.'s work comparing indices of Hadley cell strength. As a member of recent working groups on Hadley cell width, I think a paper like this is long overdue. I appreciate Pikovnik's diligence in comparing indices as a function of level and latitude and noting the impact of changing altitude or latitude on the trends from indices where the maximum value (i.e. of the stream function) is picked wherever it occurs. (An example of a paper that uses this metric is below.)

**Response:** We thank the Reviewer for their very detailed and constructive comments. We hope that all their comments and concerns are adequately addressed in our responses and that the text modifications in the revised manuscript are appropriate. Following the suggestion of Reviewer 1, we have added another metric of the HC strength based on the water vapour transport (Sohn and Park, 2013). Figures 2-5 were made more concise by showing stream-function-based metrics 100 hPa apart. Furthermore, to reduce redundancy, Figures 2 and 4 now show trends of seasonal-mean HC instead of monthly-mean HC, following the argument presented in Waugh et al. (2018). In order to follow the terminology used by other authors (e.g. Solomon et. al, 2016; David and Birner, 2018; Waugh et al., 2018), we have changed "indices" and "measures" into "metrics" throughout the text, as well as in the title.

References:

Davis, N., & Birner, T. (2017). On the Discrepancies in Tropical Belt Expansion between Reanalyses and Climate Models and among Tropical Belt Width Metrics, Journal of Climate, 30(4), 1211-1231.

Solomon, A., Polvani, L. M., Waugh, D. W., and Davis, S. M. (2016), Contrasting upper and lower atmospheric metrics of tropical expansion in the Southern Hemisphere, Geophys. Res. Lett., 43, 10,496– 10,503, doi:10.1002/2016GL070917.

Sohn, B. J., and Park, S.-C. (2010), Strengthened tropical circulations in past three decades inferred from water vapor transport, J. Geophys. Res., 115, D15112, doi:10.1029/2009JD013713.

Waugh, D. W., Grise, K. M., Seviour, W. J. M., Davis, S. M., Davis, N., Adam, O., Son, S.-W., Simpson, I. R., Staten, P. W., Maycock, A. C., Ummenhofer, C. C., Birner, T., & Ming, A. (2018). Revisiting the Relationship among Metrics of Tropical Expansion, Journal of Climate, 31(18), 7565-7581.

**Comment:** I am interested in the normal mode-based metric. I found the description in the paper inadequate; wording like "unbalanced flow" is foreign to those of us who haven't read up on the normal mode decomposition. In fact, the relationship between normal modes and the Hadley cell is not intuitively obvious; when I first read that there was a normal mode-based

metric, I thought, "this must be a usage of normal mode besides what I'm thinking." But when I read a bit of Zagar & J. Tribbia (2020), I saw that the usage of normal modes was precisely what I was used to, and "unbalanced flow" describes flow that is not in thermal wind balance. Not surprisingly, the overturning one can deduce from the unbalanced flow – see Figures 12 and 14 of Zagar and Tribbia (2020) – looks very much like the overturning one can deduce from the divergent flow – see Figure 2 of Staten et al. (2019).

**Response:** We agree with the Reviewer that the description of the new metric derived from the normal-mode function decomposition was not given enough space and was not explained in sufficient detail. This is because the new metric has yet to be fully explored and tuned, and the paper focuses on discussing already established indices. While we hope to carry out more evaluation of the new metric in the future, we expanded its definition in Section 2 (lines 110-130) and in the result sections where needed. In particular, we refer to Žagar et al. (2017), who discuss the spectrum of the global flows projecting on the inertia-gravity eigensolutions of linearized primitive equations. Their figure 15 is enclosed here as Fig. 1. They distinguish three regimes projecting on non-Rossby modes. The large-scale regime (zonal wavenumber k<6), where the Hadley cell and Walker circulation project, is denoted "unbalanced circulation". Note however that Fig. 1 does not include the zonal-mean state, as k=0 is commonly not a part of the log-log energy spectrum. But its energy would be a continuation of the power-law of -1 associated with the large-scale flow.

[Figure]

Figure 1: Three components of the circulation projecting on the inertia-gravity modes. Circulation projecting on inertia-gravity eigensolution of the linearized primitive equations at large scales is termed "unbalanced circulation". In other words, this is circulation made of non-Rossby modes. From Žagar et al. (2017, J. Atmos. Sci.), their figure 15.

Furthermore, we have added the description of the "unbalanced" circulation at multiple instances within the manuscript, including in the Introduction (lines 54-55): "Here the term `unbalanced' denotes circulation that projects on the inertia-gravity (or non-Rossby) eigensolutions of the primitive equations (Kasahara and Qian, 2000)", and in lines 110-112: "Unbalanced circulation is derived using the normal-mode function decomposition and it corresponds to the circulation projecting onto the inertia-gravity eigensolutions (or IG modes) of the linearized primitive equations (Žagar et al., 2017)."

References:

Žagar, N., Jelić, D., Blaauw, M., & Bechtold, P. (2017). Energy Spectra and Inertia–Gravity Waves in Global Analyses, Journal of the Atmospheric Sciences, 74(8), 2447-2466

**Comment:** This regionality may help explain the shortcomings of metric 8 and its lack of correlation with the other metrics. I noticed that metric 8 uses only the energy from the k=0 unbalanced flow mode. This makes sense in as much as the Hadley circulation is zonally uniform. But in reality, the Hadley cell has centers of action, and there are even longitudes with counter-Hadley-cell-wise circulations. Much of the activity of interest when we study the Hadley circulation may be regionally focused. Perhaps the observed regional variations that are changing the zonal ψ-based metric are simply not captured by the changing energy of the k=0 unbalanced mode. I wonder whether including the energy from modes with k<=4 (in the tropics and subtropics) into a metric like metric 8 would help to capture the kinds of variations that are producing the temporal variations in ψ-based metrics.

It must be a little disappointing to create such an interesting and seemingly holistic metric…only for it to apparently underperform. It is a credit to the authors that they acknowledge its underperformance in this case. I can't help but wonder if there's more to it, though.

**Response:** We agree with the Reviewer that longitudinal inhomogeneity is an important characteristic of the Hadley circulation. However, all metrics of the global Hadley cell strength perform zonal-averaging by definition. Thus, we also opted for k=0. Besides, the zonal-mean state (k=0) modes also contain the majority of the total energy (kinetic plus available potential). Adding higher zonal wavenumbers (k=1,2,3,4 or even all zonal wavenumbers) barely affects the metric's variability and trend, as shown in Fig. 1 below (lines denoting IG (k=0) and IG (k<=4) largely overlap). This suggests that regional effects do not explain the discrepancy between the ψ-based metrics and the total unbalanced k=0 energy metric.

By default, our new metric based on the unbalanced circulation energy does not include the MRG waves that are obtained as the n=0 balanced mode by the numerical procedure of the K&S (1985) used by MODES. In principle, we would like to have it included among unbalanced or non-Rossby modes. We have tested whether adding the complete spectrum of MRG waves to the original metric affects the trends. Furthermore, we have also computed the complete zonal wavenumber spectrum of unbalanced circulation with and without the Kelvin waves, which

describe a large part of the tropical zonal circulation. None of that changed the variability or trends significantly. This may be because we use monthly mean fields to compute the energy metric even though the predominant variability in both the Kelvin and MRG waves is in intra-monthly periods, as can be seen in their real-time diagnostics http://modes.cen.uni-hamburg.de. However, using daily data is computationally much more expensive, and thus not used here.

For the above reasons, we opted to continue using our original metric, the total energy contained in the zonal-mean unbalanced circulation, and we stress that a lack of correlations with other metrics does not necessarily reflect a discrepancy in physical mechanisms behind the processes making the Hadley cell and its trends.

We also note that the energy metric contains more than just the meridional wind signals, as the unbalanced circulation also represents the zonal wind and geopotential fields. Using the daily data (instead of monthly) and other refinements are necessary to improve our energy metric.

[Figure]

Figure 2: Sensitivity of the energy-based HC metric to the choice of zonal scales and inclusion of the Kelvin and MRG modes. Different lines are explained in the legend: IG refers to all inertia-gravity modes and no MRG wave, the inclusion of the Kelvin and MRG is indicated with + and -, and k stands for the zonal wavenumber.

**Comment:** Another outgrowth of Hadley cell width working group efforts that may be of use for the authors was the TropD software written by Ori Adam (see Adam et al., 2018). This software is built for detecting just the kind of metrics Pikovnik et al. use, and does so in a careful manner. I recommend trying the code and verifying that the metrics (or a subset of them) calculated by Pikovnik match those by Adam's software. The detection of extrema is particularly thoughtfully handled in TropD. It would also ensure that your results are intercomparible with those of other recent papers, such as Menzel & Waugh (2019).

Menzel & Waugh's work is worth citing, I think, even though it is nominally about the subtropical jet, rather than Hadley cell intensity. In it, they find that the subtropical jet position is more closely related to Hadley cell intensity than Hadley cell width (and that Hadley cell width is more closely related to the speed of the subtropical jet than to its position). I think this could be cited in the introduction section, as a bit of motivation for studying Hadley cell intensity, as well as another example of a paper that uses metric 1. Also, it is interesting to note that the intensity of the zonal mean overturning (which is mathematically equivalent, I think, to the zonal mean of the overturning due to the "unbalanced" flow) would be so highly correlated with the subtropical jet, which is thought of as being in thermal wind balance.

**Response:** We thank the Reviewer for the suggestion. We have verified our results with TropD software by Adam et al. (2018). The results are shown in Fig. 3 below and show an excellent agreement with that of TropD software for large n, i.e. for the case with no smoothing applied (Adam et al., Section 2.3, their Eq. 1). In particular, this paper and Waugh et al. (2018), are very relevant as they show an extensive comparison of different metrics of HC width, thus we mentioned them in the Introduction: "Lately, multiple studies have compared different metrics of the tropical expansion (Solomon et al., 2016; Adam et al., 2018; Waugh et al., 2018). On contrary, the Hadley circulation strength has only been compared between different reanalyses and climate models (e.g. Stachnik et al., 2011; Chemke and Polvani, 2019), whereas no study (to our knowledge) has yet compared the metrics of the HC strength in the same dataset."

[Figure]

[Figure]

Fig. 3. Comparison of latitudes, where maximums of different HC strength metrics are obtained; produced by our methods (full lines) vs. TropD software (dashed lines) by Adam et al., 2018. Note that the dashed lines, obtained by TropD software, are increased by +0.2° for easier verification.

**Comment:** I am concerned about the use of the Mann-Kendall test for significance. This test requires data to have no autocorrelation. The authors say they use a "modified" version, but the paper does not say how or why it was modified, nor does it address the issue of auto-correlation in the time series. Figure 1 has large regions of weak signal (for example, during March in the lower troposphere over the SH) that are marked as significant. Much of this region is not marked as significant in ERA-Interim (Figure A1). Of course, differences in the trend from one dataset to another do not imply that the trend in one dataset is not significantly significant in that dataset. But perhaps the calculation of significance needs to be handled more circumspectly. That said, I appreciated how careful the authors were to distinguish between a trend that happens to be in the time series and trend related to forced climate change.

**Response:** We agree with the Reviewer that the "modified" MK test should have been explained better in the original manuscript. Indeed, we used the modified MK test using the pre-whitening method (Yue and Wang, 2002) that accounts for the autocorrelation of the time series. Thus, we have updated the text and changed the citation (lines 144-146): "The trends are considered significant if they pass the 95% threshold of the modified Mann-Kendall test using the trend-free pre-whitening method to eliminate the impact of serial autocorrelation (Yue and Wang, 2002)."

Note that we have also experimented with other derivations of Mann-Kendall test that account for autocorrelation, e.g. Hamed and Rao, 1998, and Yue and Wang, 2004. All tests yielded similar results on the significance of the trends.

Hamed, K. H., & Rao, A. R. (1998). A modified Mann-Kendall trend test for autocorrelated data. Journal of hydrology, 204(1-4), 182-196. doi:10.1016/S0022-1694(97)00125-X

Yue, S., & Wang, C. Y. (2002). Applicability of prewhitening to eliminate the influence of serial correlation on the Mann-Kendall test. Water resources research, 38(6), 4-1. doi:10.1029/2001WR000861

Yue, S., & Wang, C. (2004). The Mann-Kendall test modified by effective sample size to detect trend in serially correlated hydrological series. Water resources management, 18(3), 201-218. doi:10.1023/B:WARM.0000043140.61082.60

Little grammatical errors, odd wording, and writing problems are scattered through the paper. I'll mention just a couple, but the paper deserves some careful editing to fix these problems.

We have made a substantial effort (to the best of our ability) to improve the language in the revised manuscript.

The rest of my suggestions are mainly editorial.

Line 28: "minimum pressure velocity" is mathematically correct but also a bit confusing when what is being described is maximum ascent.
We appreciate the suggestion and have corrected the text accordingly.

Line 30: "the studies" is better as "studies" here, since a specific group of studies has not been referred to.
We have changed "the studies" to "studies" as the reviewer suggested.

Line 42: I can see why this sentence about Nguyen et al. appears in the same paragraph as the preceding list, but the topic of the sentence feels like it belongs in a new paragraph. It's also a little disappointing that the results from Nguyen et al. aren't described here; all that is said is that

they have relevant results. I suggest making this a new paragraph, and continuing by mentioning their relevant results. (And if they're not really relevant, the sentence could perhaps be deleted.)

We have decided to remove the statement that the metric by Nguyen et al. is the only study that addresses the vertical inhomogeneity of the HC strength and its trends. Besides, the pros and cons of different metrics are described in Section 2.2 below their definitions (lines 131-140).

Line 46: "we assess how sensitive are the trends" is awkward. "we assess how sensitive the trends are" is better.

We have reformulated the sentence as (lines 48-50): "For example, we assess the sensitivity of the trends derived from metrics based on the latitude-pressure stream-function profile (\ref{eq:hc_strength}) to the choice of the pressure level".

Line 50: It is taken for granted here that readers will understand what the "unbalanced" global circulation is, even though this is not a phrase

See also a response to Major comment 1. We have provided an additional explanation at multiple instances within the manuscript, including in the Introduction (lines 54-55): "Here the term `unbalanced' denotes circulation that projects on the inertia-gravity (or non-Rossby) eigensolutions of the primitive equations (Kasahara and Qian, 2000)", and in lines 110-112: "Unbalanced circulation is derived using the normal-mode function decomposition and it corresponds to the circulation projecting onto the inertia-gravity eigensolutions (or IG modes) of the linearized primitive equations (Žagar et al., 2017)."

Line 60: While details are fine to be left in citations, some description or summary for the reader would be appreciated

We rearranged the data section, skipped the references to normal modes decomposition software, and rather expanded the description in section "Metrics of Hadley cell strength".

Line 69: "an extend" should be "the extent"
Corrected.

Line 70: "a part of the 40-year trends in the HC strength may be due to the multi-decadal variability." This is (virtually) given. Any time series can be decomposed into a trend + variability + error/noise. Say something more specific, or get rid of the sentence.

This should be true, but unfortunately, we have seen many manuscripts claiming a trend (e.g. climate change-related or model-error-related/bias-related), completely forgetting about internal variability. We merely want to point out that any 40-year trend is a combination of different factors, which we showed in a separate study as well (Zaplotnik et al., in review at J. Clim.). We think that a statement like this should be present to avoid misinterpretations of the trend by readers.

Line 74: I don't know if "feature" is the right word. It sounds like you are going to describe a particular anomaly. It might be better to say, simply, "Note that trends in…"
Corrected as suggested by the reviewer.

Line 81–82: as I mentioned above, in addition to the reliability of some climate model trends being called into question, I would question the reliability of the statistical significance test used here.
This was some uncareful wording on our side. We do not question the "climate model trends", but rather only the trends derived from the point metrics of HC intensity. We have thus removed the word "climate" from the sentence: "The presence of the inhomogeneities in the HC trends raises a question about the reliability of some of the trends derived from point metrics of the HC strength." See also related answers on the statistical significance tests above.

Lines 145–147: Of course the same feature (namely climatology) can be seen in Figure 1: Figure 1 is the climatology. Why not say, "Normalization accounts for some of these differences; normalized results are discussed near the end of Section 3.2 (see Figure 4)."
We thank the reviewer for an excellent suggestion. We have corrected the text accordingly.

Line 164: I don't know that "spurious" is the right word to describe point-wise trends. Maybe "non-representative" or "isolated." The point is not that the trends are wrong, but that they are probably not what the authors or readers are interested in.
We have removed the sentence in the revised manuscript, as the monthly-mean trends are replaced by seasonal-mean trends, where the described feature is not observed to such extent.

Line 175: "the measure of average HC strength" is vague. If it is a monthly average, any of these ψ-based metrics could be described this way.
Response: We agree that the terminology was dubious. Thus, we have changed the "average HC strength" to the "spatially-averaged HC strength" here and elsewhere in the text.

Line 243: "Insignificant correlations are not surprising as this index is largely different from all other indices." This is not the most satisfying or meaningful sentence. What does largely different mean? If the unbalanced flow is thought to be largely related to Hadley cell-wise circulation, isn't a higher correlation expected? I can't imagine the authors went through the effort to define this metric if they expected the correlations to be insignificant.
**Response:** We agree with the Reviewer that the sentence is unclear and it has been removed. There are multiple reasons for the lack of correlation. Figure 5 shows that metrics based on the same variable are correlated well; i.e. all stream function metrics are well correlated, including the new measure - spatially averaged stream function. The correlation of stream function metrics with the velocity potential and vertical velocity drops down to about 0.3-0.5. It is not surprising that a correlation with a very different metric based on a quadratic quantity that is associated with the stream function through the square of its vertical gradient is poor. It is hard to bring forward physical arguments behind the \Psi-\omega correlation being about 0.4, and we do not have an explanation for the physical nature of correlations between the unbalanced energy and the stream function.

Line 246: "unimportant for the overall signal, but it may be important for the trends". What is the difference between the signal and the trends? Often "signal" is used to describe "trends" and "noise", is used to describe year-to-year or decade-to-decade "variability." Did the authors mean "signal" or "climatology"?

We have now modified the text in lines 282-283 (former line 246) as: "The part of $I\_M$ from the extra-tropics and the stratosphere is however arguably unimportant for the overall magnitude of the metric."

We have also alleviated the use of signal and noise in the revised manuscript to avoid misinterpretation.

Lines 250–252: Does this result imply that metric 8 is going to be sensitive to increasing tropopause height?

Metric 9 (former metric 8) as defined in the manuscript is not directly sensitive to increasing tropopause height, since the energy of the unbalanced motions remains steady. However, by imposing an artificial lid (e.g. at 100 hPa) like in the described experiment, the metric becomes sensitive to HC strength.

Line 265: "This was made evident by a new HC strength measure." It is not clear what was made evident, or how it was made evident.
We agree with the Reviewer that the text is dubious. It is a remnant of an earlier manuscript version. Thus, we decided to remove the sentence as well as the preceding sentence.

Line 285: Based on my working group experience on Hadley cell width, I am skeptical of the idea of a "unified index." Different metrics for Hadley cell width are of interest to different people for different reasons, even if their trends differ. If we created a unified index, there would be information missing from that index for each group. A unified index for Hadley cell strength would not capture simultaneously the differences between hemispheres, between regions, and in the deep tropical upwelling. Upper tropospheric circulation may be of interest to people studying the stratosphere or the tropical tropopause region, while mass is concentrated in the lower troposphere, so mass-weighted measures will leave them out.

We fully agree with the Reviewer on this one. What we meant was that studies like the IPCC report and similar other reports/studies that talk about a general strengthening/weakening of the HC should start using a "unified" HC metric, or to be more specific about why a certain metric is used over the other metric. Similarly, other studies should be specific about their use of the HC strength, and why they use it. As we point out earlier (lines 319-323) in the text, the use of the indices will depend on what we want to study (exactly what Reviewer suggests here). We have now rephrased the sentence to be more specific as to what we mean. See l. 319-326.

---

## Author Comment (AC3)

Review #1 of 'Indices of the Hadley circulation strength and associated circulation trends' by Pikovnik et al.

**General comments:**

This manuscript compares 8 measures of the HC strength derived from ERA5 and ERA-interim reanalysis datasets. Their main findings are that measures based on a single vertical level are more subject to uncertainty and inhomogeneity while measures based on spatial average or integration are more robust. They concluded that the measure of the average HC strength is best suited for studying variability and trends.

The comparison is interesting and the conclusions are pertinent. However, 7 out of the 8 measures are derived directly or indirectly from the zonal mean stream function, which explains the high correlations between measures. The one measure not derived from the stream function is deemed inadequate for this study and needs further refinement. Perhaps it would have been important to compare independent measures of the HC strength and quantify their relative relevance rather than the 7 measures proposed here as it is intuitive that capturing the HC by taking into account both its meridional and vertical extent would be more robust than from a single location.

We thank the Reviewer for their constructive comments. We hope that all their comments and concerns are adequately addressed in our responses and that the text modifications in the revised manuscript are appropriate. Figures 2-5 were made more concise by showing stream-function-based metrics 100 hPa apart. Furthermore, to reduce redundancy, Figures 2 and 4 now show trends of seasonal-mean HC instead of monthly-mean HC, following the argument presented in Waugh et al. (2018). In order to follow the terminology used by other authors (e.g. Solomon et. al, 2016; David and Briner, 2018; Waugh et al., 2018), we have changed "indices" and "measures" into "metrics" throughout the text, as well as in the title.

As the Reviewer points out, our suggestion that the metrics of the HC strength that take into account both the meridional and vertical extent of the global HC are overall better indices than the HC metrics based on local values may be intuitive. The time series of the stream function-based indices are aligned (Fig. 3 in the revised paper) and highly correlated (Fig. 5), however, the differences become important when one computes the HC trends and quantifies their uncertainties. This is the first lesson from our comparison of independent metrics of the HC strength. We applied the metrics from previous studies to bring our results into the context of the reported trends in the HC strength. We agree that the trends based on other independent metrics could be explored, such as the water vapor flow of Sohn and Park (2013). This metric is included in the revised manuscript (see metric nr. 8 in Section 2.2, lines 102-109). We have searched the peer-reviewed literature for other metrics and have not found any, so to our knowledge, these are the only metrics that have been used in the past.

It is unclear how to quantify the relative relevance of metrics if we do not have a reference (or generally agreed) HC strength metric to which various other metrics can be compared. It is in this context that we introduced a new, energy-based integral metric of the HC strength. The unbalanced energy of the zonal mean circulation is straightforward to derive for gridded datasets and it includes all 3 spatial dimensions of the unbalanced circulation. We are not sure what exactly the reviewer finds confusing about the new energy metric, but the metric is discussed in more detail in the revised manuscript. It is different from the stream function, but also from the omega-based and velocity-potential-based metrics, and more research can be done to refine it, especially to differentiate between the northern and southern branches of the Hadley cell. Our figures A2 and A3 suggest that the global unbalanced circulation is an adequate description of the HC, implying that the associated total energy is an adequate description of the HC strength. We believe that it is a suitable metric also for an intercomparison of reanalyses and climate models analyzed in terms of the normal-mode functions.

Davis, N., & Birner, T. (2017). On the Discrepancies in Tropical Belt Expansion between Reanalyses and Climate Models and among Tropical Belt Width Metrics, Journal of Climate, 30(4), 1211-1231.

Solomon, A., Polvani, L. M., Waugh, D. W., and Davis, S. M. (2016), Contrasting upper and lower atmospheric metrics of tropical expansion in the Southern Hemisphere, Geophys. Res. Lett., 43, 10,496– 10,503, doi:10.1002/2016GL070917.

Sohn, B. J., and Park, S.-C. (2010), Strengthened tropical circulations in past three decades inferred from water vapor transport, J. Geophys. Res., 115, D15112, doi:10.1029/2009JD013713.

Waugh, D. W., Grise, K. M., Seviour, W. J. M., Davis, S. M., Davis, N., Adam, O., Son, S.-W., Simpson, I. R., Staten, P. W., Maycock, A. C., Ummenhofer, C. C., Birner, T., & Ming, A. (2018). Revisiting the Relationship among Metrics of Tropical Expansion, Journal of Climate, 31(18), 7565-7581.

**Specific comments:**

I find it strange to choose 2 versions of the ECMWF reanalysis instead of 2 new-generation products such as ERA5 and CFSR for a more independent comparison. It's been reported that ERA5 is an improved version of ERAI with many significant fixed errors therefore the discrepancies found by the authors may be attributed to those improvements.

In their specific comment, the Reviewer points out that ERA5 is a more advanced and therefore more reliable reanalysis dataset than ERA-Interim. We could not agree more and we emphasize this out in the revised paper (lines 68-76). Although relatively few evaluations of the CFSR have been conducted and thus its performance is not well-known, we believe that CFSR is much more advanced than the NCEP-NCAR reanalyses. Yet, many researchers would argue that even the NCEP-NCAR reanalysis suffices for the description of the large-scale circulation. Even

though ERA5 is available, many scientists rely on ERA-Interim, and precisely a comparison of tropical aspects in the ERA5 and ERA-Interim, which have been the subject of several recent papers, motivated our study initially. Even though the tropics remain the region with the largest analysis uncertainties (e.g. Žagar et al., 2020, J. Clim), the four modern reanalyses (ERA5, ERA-Interim, JRA55, and MERRA) agree relatively well regarding the large-scale tropical circulation. The total energy of the zonal-mean unbalanced flow shows positive trends in both ERA5 and ERA-Interim, although weaker in the former, further suggesting consistency between the two reanalyses.

Note, however, that the only aim of choosing another reanalysis besides ERA5 was to show that the sensitivity of HC strength trends to the choice of the metric is not an isolated feature of a particular (e.g. ERA5) reanalysis, as stated in lines 299-300. A detailed comparison of the HC strength trends in various reanalyses and the sources of their differences are beyond the scope of this study.

Žagar, N., Zaplotnik, Ž., & Karami, K. (2020). Atmospheric Subseasonal Variability and Circulation Regimes: Spectra, Trends, and Uncertainties, Journal of Climate, 33(21), 9375-9390.

**Technical corrections:**

L46: suggest replace '…are the trend… the pressure level.' by '…the trend… the pressure level are.'
Corrected as suggested.

Section 2.2: this should go in the result section, not in the methods section
We followed the Reviewers suggestion.

L177: suggest remove 'also'
Corrected.

L182: what do the authors mean by 'merely showcase'?
We have corrected it to a more neutral form: "Figs. 2, A3 reflect the stronger year-to-year variability of seasonal means (compared with year-to-year variability of annual means),…"

---

## Author Response (AR2)

Review #2 of 'Indices of the Hadley circulation strength and associated circulation trends' by Pikovnik et al.

We thank the Reviewers for their suggestions. Answers to their comments are indicated in blue font.

**Reviewer 2**

The authors have made considerable effort to respond to or address the criticisms and suggestions of the reviewers. I am satisfied with the paper in its current form.

The word "however" on line 337 should be flanked by commas.

We have double-checked all "however" and added commas where relevant (incl. the reviewer's suggestion).

**Reviewer 3**

The authors should compare their findings on the energy-based metric with the mass transport metric found in Lucas et al (Variability and changes to the mean meridional circulation in isentropic coordinates. Clim Dyn 58, 257–276, 2022) who also found strengthening of the SH HC inferred from ERAI.

Thank you for pointing out this new study. While we acknowledge other than pressure system analysis of the mean meridional circulation including the study suggested by the Reviewer, comparing our results from the pressure system with other coordinate systems is beyond the scope of the present study. However, we list such a comparison as one possible way to expand on the present work.

In Conclusions (l. 321-324), we explicitly mention that the zonal-mean stream function can be computed as well in other coordinate systems:
"Note that evaluations of the HC strength and its trends may also profit from analyses in alternative coordinate systems, such as thermodynamic coordinates (Kjellson et al, 2014), moist isentropic coordinates (e.g. Wu et al., 2019) or dry isentropic coordinates (e.g. Lucas et al., 2021) that yield a different perspective on the mean meridional circulation."

We also mention in the Introduction (l. 47) that our focus is on pressure-coordinate metrics which are used most widely:
"We assess the sensitivity of the trends derived from the stream-function (1) based metrics in isobaric coordinates (Eq. 1) to the choice of the pressure level."

Note that we have not found significant strengthening of SH HC in ERA-Interim using any of our metrics besides water vapour transport metric (8), as indicated in Fig. 4d. While the energy-based metric shows the strengthening of the overall global zonal-mean unbalanced circulation, it cannot distinguish between the SH HC and NH HC. The apparent strengthening of the SHC in ERA-Interim might be due to the ambiguity of former Fig. 4. Thus, we have now separated the metrics describing NHC strength, SHC strength, and the total Hadley circulation strength in new Fig. 4 to avoid possible misinterpretation.

I find the figures 2 through to 5 and Table 1 hard to follow, along with the text. It would make it simpler if once each metric is defined, to refer them to their number.

We agree with the reviewer that the text should be simplified. We have thus provided more links/references to the Figures/metric numbers in the text and captions – see e.g. lines 202, 208, 217 in the track-changes file, Fig. 3 caption, etc. We have also unified the expressions for certain metrics: metric (7) is now referred to as the "average stream-function metric", whereas we refer to metric (9) simply as "unbalanced energy metric". A larger portion of Fig. 4 caption is removed, and the reader is referred to the Methodology section (2.2) and Fig. 3.

The authors concluded that metric 7 was the best choice due to their integration in both the meridional and vertical as opposed to single point or vertical integration only. While this is a valid point, the reader is left wondering about metric 8 and 9, which are the most interesting to compare with the wind-based metrics. A discussion is needed here.

We do not claim that the metric 7 is the best metric, rather that it alleviates problems of making strong conclusions based on a single point or level.

We have added a discussion. See l. 317-321:
"However, our results demonstrate that caution is needed when comparing HC trends from different studies using different metrics of the HC strength. In light of all the results, we would suggest using the average stream function as the metric of the overall HC strength whenever interested in the variability and trends in each Hadley cell separately. On the other hand, the unbalanced energy metric is a physically-sound choice for analysing the changes in the global zonal-mean circulation."

Other non-stream-function metrics, such as (5), (6) and (8) only give a partial description of the HC features (e.g. upper branch in proximity of the ascending branch, only ascending branch or only lower branch, respectively) - see discussion in lines 132-137.